# Comparative analysis of data-driven models for spatially resolved thermometry using emission spectroscopy

**Ruiyuan Kang**[1], **Dimitrios C. Kyritsis**[2], **Panos Liatsis**[3]*

**1** Directed Energy Research Center, Technology Innovation Institute, Abu Dhabi, UAE, **2** College of Technology and Design, Neom University, Neom, Saudi Arabia, **3** Department of Computer Science, Khalifa University, Abu Dhabi, UAE

* panos.liatsis@ku.ac.ae

## Abstract

A methodology is proposed, which addresses the caveat that line-of-sight emission spectroscopy presents in that it cannot provide spatially resolved temperature measurements in non-homogeneous temperature fields. The aim of this research is to explore the use of data-driven models in measuring temperature distributions in a spatially resolved manner using emission spectroscopy data. Two categories of data-driven methods are analyzed: (i) Feature engineering and classical machine learning algorithms, and (ii) end-to-end convolutional neural networks (CNN). In total, combinations of fifteen feature groups and fifteen classical machine learning models, and eleven CNN models are considered and their performances explored. The results indicate that the combination of feature engineering and machine learning provides better performance than the direct use of CNN. Notably, feature engineering, which is comprised of physics-guided transformation, signal representation-based feature extraction and Principal Component Analysis is found to be the most effective. Moreover, it is shown that when using the extracted features, the ensemble-based, light blender learning model offers the best performance with RMSE, RE, RRMSE and R values of 64.3, 0.017, 0.025 and 0.994, respectively. The proposed method, based on feature engineering and the light blender model, is capable of measuring nonuniform temperature distributions from low-resolution spectra, even when the species concentration distribution in the gas mixtures is unknown.

**Data Availability Statement:** The dataset is available at https://www.kaggle.com/datasets/ralphkang/emission-spectra. All software code is available at https://github.com/RalphKang/Spatially_temp_measurement.

## 1 Introduction

Emission Spectroscopy (ES) is a popular gas analysis technique, which relies on the processing of emitted light, when a gas mixture is interrogated via an exciting signal. Its measurement principle follows the traditional formulation of the forward model, where a mixture of chemical species, distributed in a 3D space, i.e., the cause, interacts with an exciting stimulus, and a measured spectrum signal is observed, i.e., the effect. The forward model is a complex yet deterministic function, which maps the state of the cause onto the corresponding effect. The

**Funding:** The author(s) received no specific funding for this work.

**Competing interests:** The authors have declared that no competing interests exist.

resulting spectrum encodes valuable information about the composition, spatial distribution and properties of the gas mixture. In the associated inverse problem, the underlying properties of the chemical sample that generated the measured spectrum can be recovered, by accessing the information contained within it. Specifically, the solution of the inverse problem results to knowledge of the mixture's composition, temperature, density, etc, along the line-of-sight.

Due to the ill-posed and severely rank-deficient nature of the ES inverse problem, its application is limited to homogeneous temperature/composition measurements. Instead, advanced techniques, such as tomographic spectroscopy [1] and laser-induced fluorescence [2] can be used to provide spatially resolved temperature measurements. Nonetheless, ES remains advantageous as it provides for greater convenience, simplicity, and cost-effectiveness. Indeed, the spatial information of the measurements is not lost during the ES scanning process, but rather it is encoded within the corresponding emission spectrum. Thus, reconstruction of spatially resolved temperature distribution is a challenging yet promising research direction.

The primary objective of this study is to develop and evaluate a comprehensive framework for spatially resolved temperature estimation from emission spectra, integrating feature engineering and machine learning (ML) approaches. By addressing the limitations of conventional Deep Learning (DL) [3, 4] and exploring the potential of hybrid methodologies, this research aims to enhance the predictive accuracy and reliability of measurements, ultimately contributing to advancements in various applications ranging from industrial combustion monitoring to environmental assessment. The contributions of the work are as follows:

1. To the best of the authors' knowledge, this is the first work, which systematically explores and compares the use of feature engineering and machine learning vs deep learning algorithms for spatially resolved temperature measurement estimation.

2. The top performing method involves a combination of signal representation-based feature extraction, and blending-based ensemble machine learning, achieving cutting-edge performance, which outperforms state-of-the-art approaches and deep learning algorithms.

3. It is demonstrated that the proposed approach is capable of accurately providing spatially resolved temperature estimates using lower resolution ES spectra, compared to published work, which offers an important cost advantage for real-world applications.

The manuscript is organized as follows: Section 2 provides a detailed overview of related works and state-of-the-art methodologies in emission spectroscopy and temperature estimation. Section 3 outlines the methodology, including the data collection process, feature extraction techniques, and employed machine learning models. In Section 4, the results of the experiments are presented, followed by a discussion of the findings in Section 5. Finally, Section 6 concludes the study and outlines future research directions.

## 2 Related work

Using emission spectroscopy for temperature measurement has been well explored, however, due to the ill-posed nature and high nonlinearity of the problem, utilizing emission spectroscopy for spatially resolved temperature measurement has not been extensively investigated. The primary research efforts in the temperature measurement can be respectively categorized into theoretical, optimization-based and machine learning-based methods.

### 2.1 Theoretical approaches

A classical yet practical method for measuring temperature from ES is the two-color method [5], where spectral lines with constrastive absorption characteristics are used to calculate the

intensity ratio and a lookup table is created to record the mapping between the ratio and temperature for temperature measurement. Utilizing the line broadening effect to calculate temperature has also been popular [6]. However, it is notable that classical methods assume that the gas to be measured has a uniform temperature distribution, and therefore, their use cannot be generalized in the case of spatially resolved temperature measurements.

## 2.2 Optimization-based approaches

Early works by Kim and Song [7] and Ren et al. [8] utilized regularization strategies and optimization algorithms to extract spatially resolved temperature information from line-of-sight spectra. These methods leverage forward simulation models to iteratively optimize the fit between simulated and actual spectra. Kim et al. [9] proposed a fast inversion scheme for spatial temperature measurements in large-scale furnaces, relying on spectral remote sensing (SRS) technology. Ren et al. [7] combined Levenberg-Marquardt optimization and Tikhonov regularization to recover the temperature profiles of $CO_2$. While these methods can provide high accuracy under ideal conditions, their performance is highly sensitive to the initial guess of the temperature distribution and the selection of hyperparameters, impacting on their robustness in the case of complex scenarios.

## 2.3 Machine learning-based approaches

Recent advancements shifted the research focus towards machine learning [10–12], which offers the advantage of learning from data without requiring explicit physics-based models. However, these technologies mainly focused on the detection [13], average composition estimation [14], and reconstruction of hyperspectral images [15, 16]. In addition, classical machine learning algorithms such as Principal Component Analysis (PCA) and multi-linear regression [17], Convolutional Neural Networks [18], and transfer learning [19] were previously explored in average temperature measurement estimation. In the context of the inverse problem of line-of-sight emission spectroscopy, studies by Cięszczyk et al. [20] and Ren et al. [21] utilized multilayer perceptrons (MLP) to estimate spatial temperature distributions. Cięszczyk et al. [20] developed a model based on the ratios of radiation intensities at multiple $CO_2$ lines, which was applied to reconstruct plume temperature distributions. Ren et al. [1] explored a complex MLP architecture with three hidden layers, which was used to retrieve temperature and concentration profiles from infrared emission spectra. While effective for relatively simple temperature distributions, these models often struggled to generalize to more complex and non-uniform profiles, due to their reliance on the raw spectral data, i.e., they do not exploit the benefits of feature engineering.

## 2.4 Comparative analysis of feature engineering

A limited number of studies have systematically explored the role of feature engineering in enhancing model performance for spatial temperature measurements. The work by Wei et al. [15] introduced deep neural networks (DNNs) [3] combined with feature extraction techniques to estimate temperature distributions in methane flames. However, their feature extraction strategy is limited to standard statistical descriptors, which lack the ability to fully exploit the underlying structure of the spectral data. In contrast to previous efforts, in this research, a comprehensive feature engineering methodology is introduced, including signal representation and physics-guided transformations, which significantly improves the model's ability in handling complex temperature profiles.

Table 1 provides a comparison of previous research efforts in this field, summarizing the methodologies, datasets, applied models, and key findings. Despite advances in both

**Table 1. Summary of related work.**

| Study | Methodology | Data | Models | Findings | Disadvantages |
|---|---|---|---|---|---|
| Parameswaran et al. (2014) [5] | Two-color method | Flame emissions spectroscopy | Two-color method | high consistency for uniform temperature profiles | Unsuitable for nonuniform profiles |
| Yubero et al. (2013) [6] | Line-broadening-effect-based method | Plasma torch emissions spectroscopy | Line-broadening-effect-based method | Applied in Plasma torch | Unsuitable for nonuniform profiles |
| Park et al. (2021) [17] | PCA and multi-linear regression | Nitrogen plasma | Multi-linear regression | Effective for uniform temperature profiles. | Unsuitable for nonuniform profiles |
| Kim et al. (2023) [18] | Deep Neural Networks | Nitrogen plasma | Convolutional Neural Network | Effective for uniform temperature profiles. | Unsuitable for nonuniform profiles |
| Yi et al. (2021) [19] | Deep Neural Networks | Nitrogen plasma | Transfer Learning | Alleviate data collection requirements | Unsuitable for nonuniform profile |
| Kim & Song (2005) [7] | Regularization and optimization-based inverse method | Simulated furnace spectra | Tikhonov regularization, SRS inversion | Achieved high accuracy in large-scale furnace temperature profiles. | Sensitive to initialization and hyperparameter choice. |
| Ren et al. (2014) [8] | Levenberg-Marquardt optimization with regularization | $CO_2$ spectral data | Levenberg-Marquardt optimization | Effective for uniform temperature profiles. | Limited scalability to complex, non-uniform profiles. |
| Cięszczyk et al. (2015) [20] | Ratio of radiation intensities and MLP | $CO_2$ line intensities | MLP (single hidden layer) | Good performance in uniform plume temperature estimation. | Struggles with complex and non-Gaussian temperature distributions |
| Ren et al. (2019) [21] | Infrared emission spectra and MLP with three hidden layers | Simulated temperature profiles | MLP (three hidden layers) | Demonstrated feasibility of MLP for spatial temperature reconstruction | Requires substantial tuning, while feature extraction was minimal |
| Wei et al. (2020) [15] | Deep neural networks with naive feature extraction | Methane flame spectra | Deep neural networks | Achieved good accuracy in simple profiles; first study to apply feature engineering in ES. | Limited feature set, lacks robustness in complex scenarios. |

optimization-based and ML methods for temperature estimation from emission spectroscopy, significant gaps remain. Optimization-based approaches are often highly sensitive to the initial conditions and choice of hyperparameters, limiting their robustness in complex scenarios. ML models, while promising, have primarily focused on the treatment of simple temperature distributions and struggled to generalize to more complex temperature profiles due to insufficient feature engineering. Furthermore, a handful of studies explored the role of advanced feature extraction techniques to enhance model performance, particularly, in handling non-uniform temperature fields. This study addresses these gaps by incorporating comprehensive feature engineering with machine learning to improve accuracy in complex temperature estimations.

## 3 Data generation and system methodology

The work plan consists of two phases: (a) data generation, and (b) system methodology. Due to the unavailability of open-source datasets, the initial phase of this study involved the generation of a suitable dataset. In this step, a high-fidelity spectral forward simulation model was build to generate emission spectra from temperature and mole fraction distributions. Following the dataset generation, temperature distribution reconstruction was performed, under the uncertainty of mole fraction fluctuation. Fig 1 provides a graphical depiction of the relationship between the two work phases. Sec. 3.1 will introduce the formulation of the forward modeling in emission spectroscopy, followed by spectral acquisition. In Sec. 3.2, the data-driven methodologies explored in this study will be introduced.

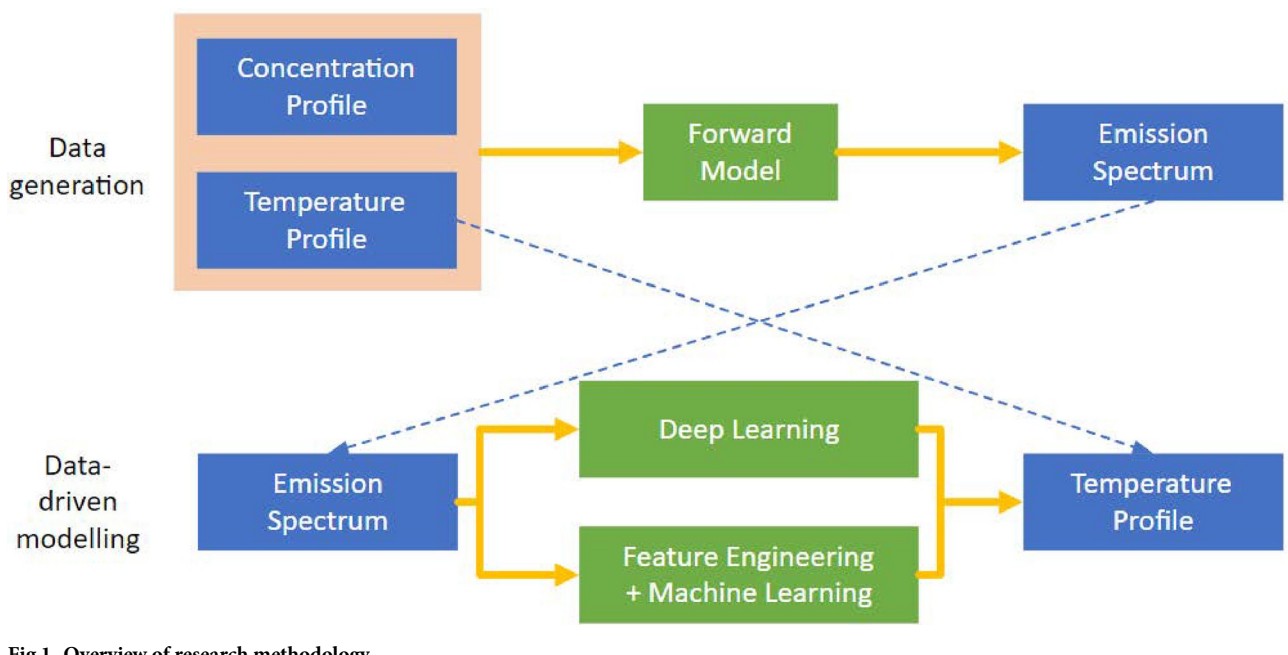

**Fig 1. Overview of research methodology.**

## 3.1 Modeling and data generation

This section primarily covers the data preparation process for data-driven modeling. Sec. 3.1.1 discusses the development of the physical forward model for spectra generation from temperature and mole fraction profiles. In Sec. 3.1.2, the physical model is applied in synthesizing the spectral data for use in the data-driven system development and testing.

**3.1.1 Physical forward model.** A forward model was constructed to simulate the emission spectra from the information of mole fraction and temperature distribution, based on the well-established HITEMP2010 database [22], and HAPI [23]. The Finite Element Method (FEM) was applied to discretize the continuous temperature and mole fraction distribution along the light path into a number of segments, each characterized by uniform temperature and mole fraction. A visual representation of the path of light passing through a flame is given in Fig 2.

According to spectroscopy theory [24], the intensity at the exit of a segment is the sum of the emission of the current segment and the transmission of the radiation from the previous segments:

$$I_{v,o} = \epsilon_v I_{v,B} + t_v I_{v,i} \tag{1}$$

where $I$, $\epsilon$, $t$ are the intensity, emissivity, and transmissivity, respectively. Subscripts $v,B,i$, and $o$ relate to the frequency, Black-body radiance, incident, and outlet nodes of the segment, respectively.

According to the Beer-Lambert law [24], transmissivity can be expressed as:

$$t_v = exp(-k_v l) \tag{2}$$

where $k_v$ is the absorption coefficient ($cm^{-1}$), and $l$ is the length of the light path segment (cm).

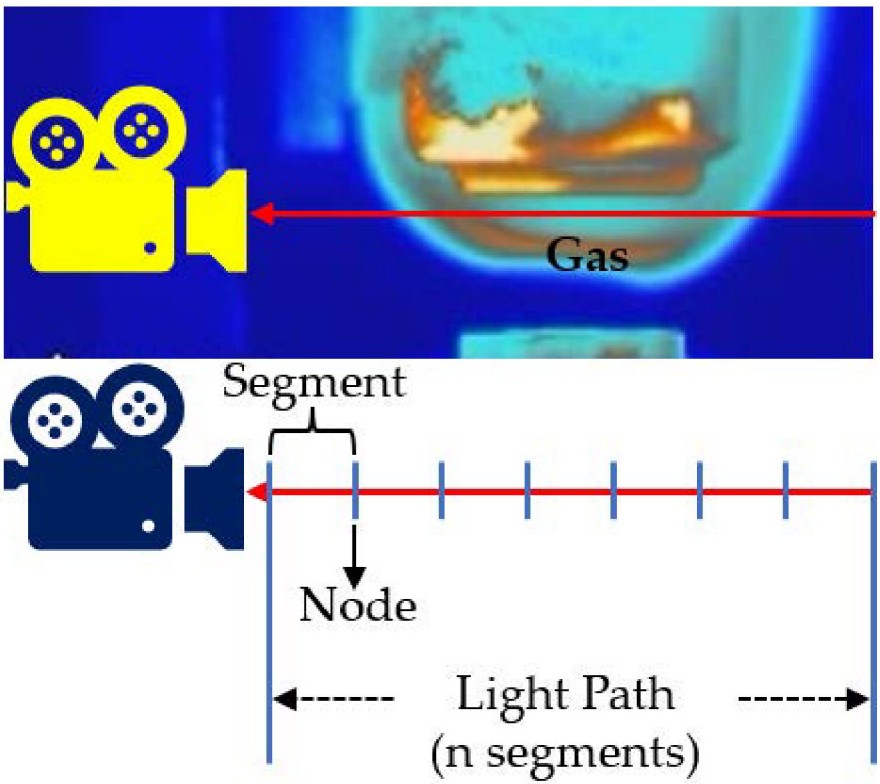

**Fig 2. Schematic diagram of light path division.**

The sum of transmissivity $t$ and emissivity $\epsilon$ is equal to 1, when scattering and reflection are neglectable, i.e.,

$$t_v + \epsilon_v = 1 \tag{3}$$

Next, it is assumed that the gas cloud is at thermal equilibrium. By following Kirchhoff's law of thermal radiation [24], i.e., emissivity equals to absorptivity, and substituting Eqs 2 and 3 into Eq 1, we obtain:

$$I_{v,o} = (1 - exp(-k_v l))I_{v,B} + exp(-k_v l)I_{v,i} \tag{4}$$

Moreover, the black-body radiance can be modeled through Plank's law [25]:

$$I_{v,B} = \frac{2hv^3}{c^2 \left( exp\left( \frac{hv}{k_B T} \right) - 1 \right)} \tag{5}$$

where $h$ is the Plank constant, $k_B$ is the Boltzmann constant, $c$ is the light speed, and $T$ is the temperature. Next, the absorption coefficient, $k_v$, of gas species $j$ is given by:

$$k_{v,j} = s_{v,j}(T)\phi_v(T,p)\frac{pX_j}{k_B T} \tag{6}$$

where $s$ is the line intensity per molecule $((cm)^{-1}/(molecule * cm^{-2}))$, which is a function of temperature, $\phi_v$ is the Voigt profile [26], which is a function of both pressure and temperature, $p$ is the local pressure $(Pa)$, and $X$ is the mole fraction.

The total absorption coefficient can be simplified as the sum of the absorption coefficients of each species:

$$k_v = \sum k_{v,j} \tag{7}$$

Despite that high resolution spectra can capture precise information with respect to temperature and mole fraction distributions, low-resolution spectra, typically affected by blurring due to line broadening and the slit function of the instrument, are the norm in engineering applications. A triangular function was used to simulate the instrument slit function as follows:

$$B(x) = \begin{cases} \dfrac{1 - \dfrac{abs(v - v_c)}{w}}{w}, & abs(v - v_c) \leq w \\[4mm] 0, & abs(v - v_c) > w \end{cases} \tag{8}$$

where $w$ is the wing of the triangular function, set to 10 $cm^{-1}$, and $v_c$ is the center of the triangular function. In this work, the resolution of spectrum is set to 4 $cm^{-1}$.

By considering the aforementioned formulae, a number of factors can be identified, which hinder retrieval of spatially resolved temperature measurements. First, as observed in Eq 6, both temperature and species concentration affect the appearance of spectra. Thus, lack of prior information in regards to the species concentration increases the difficulty in retrieving the varying temperature profile along the light path. Second, as shown in Eq 7, in the case of gas species mixtures, the total absorption coefficient is affected by the spectral characteristics of all species, thus, posing a further complication in estimating the temperature measurement information. Third, blurring due to various factors, including the instrument slit function of Eq 8, reduces the quality of spectral information, thus increasing the ambiguity in temperature measurement estimation. Last but not least, temperature profile non-uniformity and lack of prior information in respect to the range of the temperature distribution complicate the extraction of patterns, governing the spectra.

**3.1.2 Spectra generation.**   In this research, we focus on the paradigm of combustion. This is a suitable scenario for spatially resolved temperature measurements using emission spectroscopy, because (1) the high temperatures of the combustion process provide sufficient emission signals, and (2) most practical combustion phenomena naturally possess nonuniform temperature distributions, which, in turn, require spatially resolved temperature measurements. During dataset generation, the conditions, which occur in the combustion process were replicated, however, more complex scenarios were also considered. This was achieved by varying each of the four parameters, which affect the appearance and quality of the spectra, subsequently affecting temperature profile estimation, namely, uncertainties in concentration, mixture of species, nonuniformities and range of temperature distribution, and spectral resolution. This methodology was followed to systematically assess the capabilities of ML/DL algorithms in tackling the spatially resolved temperature measurement estimation problem. The detailed data configuration is described in the following paragraphs.

The gases considered were CO, $CO_2$ and $H_2O$. As previously mentioned, such gas mixtures also complicate temperature measurement estimation. Following [21], the wavebands selected were in the range of 1800-2500 $cm^{-1}$, which covers the emission bands of these three substances. The light path was set to be 10 cm and divided into eleven segments, i.e., twelve nodes, with each segment being less than 1 cm in length. The relatively small segment length and the choice of the number of segments suffice to validate the feasibility of measuring spatially resolved temperature distributions. Indeed, the number of segments may vary according to

the requirements of the application. The dual-peak Gaussian function, commonly encountered in flames [27], was used as the base profile herein. The ideal dual-peak Gaussian function is given by:

$$\rho_{DG,j} = exp\left(-n\left(\frac{j-n}{4\sigma}\right)^2\right) + exp\left(-n\left(\frac{3(j-n)}{4\sigma}\right)^2\right) \tag{9}$$

where $\rho_{DG,j}$ is the density of dual-peak Gaussian profile at segment $j$, $n$ is the number of segments, $\sigma$ is the spread, which was empirically set to 16. The values of $\rho_{DG,j}$ were normalized in the range of [0, 1]. The feasible ranges of temperature and mole fraction can be obtained from the normalized density as follows:

$$T_{DG,j} = \rho_{DG,j,norm}(T_{max} - T_{min}) + T_{min} \tag{10}$$

$$X_{DG,j} = \rho_{DG,j,norm}(X_{max} - X_{min}) + X_{min} \tag{11}$$

where $\rho_{DG,j,norm}$ is the normalized basis height, $T_{DG,j}$ and $X_{DG,j}$ are, respectively, the temperature and mole fraction at segment j, $T_{max}$, $T_{min}$ and $X_{max}$, $X_{min}$ are, respectively, the maximum and minimum of the temperature range and mole fraction range. The corresponding temperature range was set to 1500-3100 K, i.e., covering a broad interval of 1600 K, while the mole fraction range was set to 0.095-0.15, with examples of ideal generated profiles, shown in Fig 3(a).

Next, the values of temperature and gas concentration in each segment were randomly varied in the ranges of ±300 K and ±0.015, respectively. Examples of temperature profiles generated by introducing such variations are shown in Fig 3(b). It is evident that such variations have a significant impact on the shape of generated profiles. For instance, see Fig 3(b), cases 1 and 2, which are generated from such profile configurations, do not exhibit the traditional shape of the dual peak Gaussian function. Rather, they resemble the shapes of trapezoidal and parabolic profiles, respectively. It should be emphasized that such profiles are also commonly observed in boundary-layer flow and flames [28, 29]. Given the process of light path discretization, a large variety of profile patterns can be generated, leading to higher spectral pattern complexity, thus increasing the difficulty in obtaining spatially resolved temperature

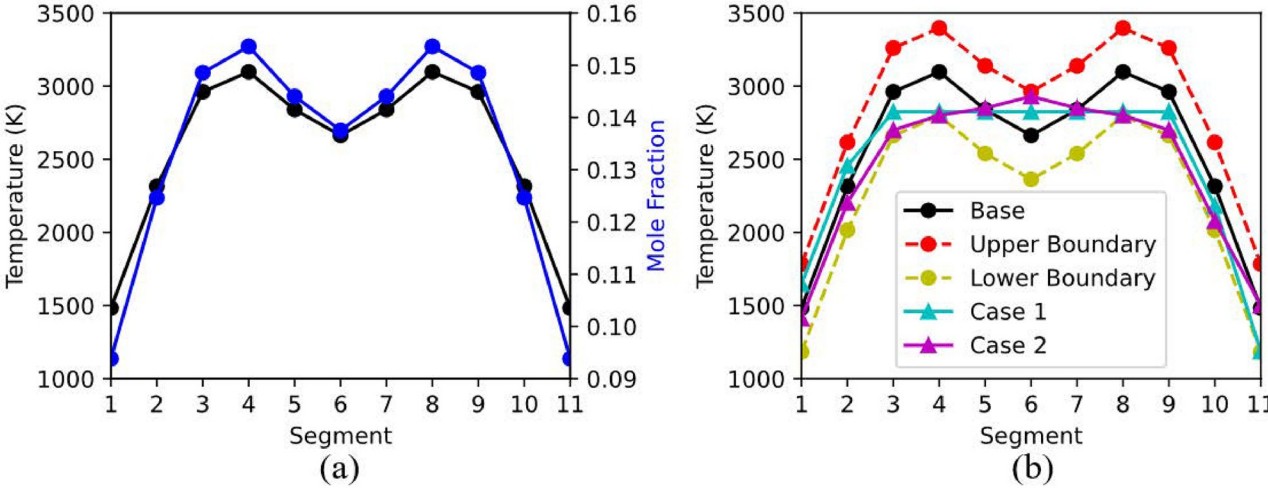

**Fig 3. Temperature and concentration profiles.** (a) Baseline– ideal profile, (b) Examples of temperature profiles, generated through random variation of the associated parameters.

measurements. In our experiments, we considered a spectral resolution of 4 $cm^{-1}$. The purpose behind this choice is that a lower resolution leads to higher spectral generation speed, and reduces instrument costs, which are often required in engineering applications. In addition, this also increases the degree of complexity of estimating temperature measurements, thus making the problem even more challenging to solve.

In total, 28,000 spectra were generated. An example distribution is shown in Fig 4. Such a low-resolution spectrum comprises of 6799 emission lines. Compared to the high-resolution spectra of 0.1 $cm^{-1}$, also shown in the Figure, it is clear that low-resolution spectra is affected by blurring, leading to loss of detail, which increases the difficulty of the reconstruction task.

The detailed description of the dataset is as follows:

1. Size and Composition: The primary dataset used for this study consists of a total of 28000 labeled emission spectra, each corresponding to a unique temperature profile. The spectra were generated using profiles of variants of the dual-peak Gaussian functions to ensure a diverse representation of temperature conditions. The dataset was divided into three subsets: training, validation, and test sets with a ratio of 70%, 15%, and 15%.

2. Diversity of Temperature Profiles: The dataset includes a wide range of temperature profiles along with a wide range of mole fraction profiles. This diversity was introduced to mimic real-world scenarios, such as non-homogeneous temperature fields in industrial furnaces

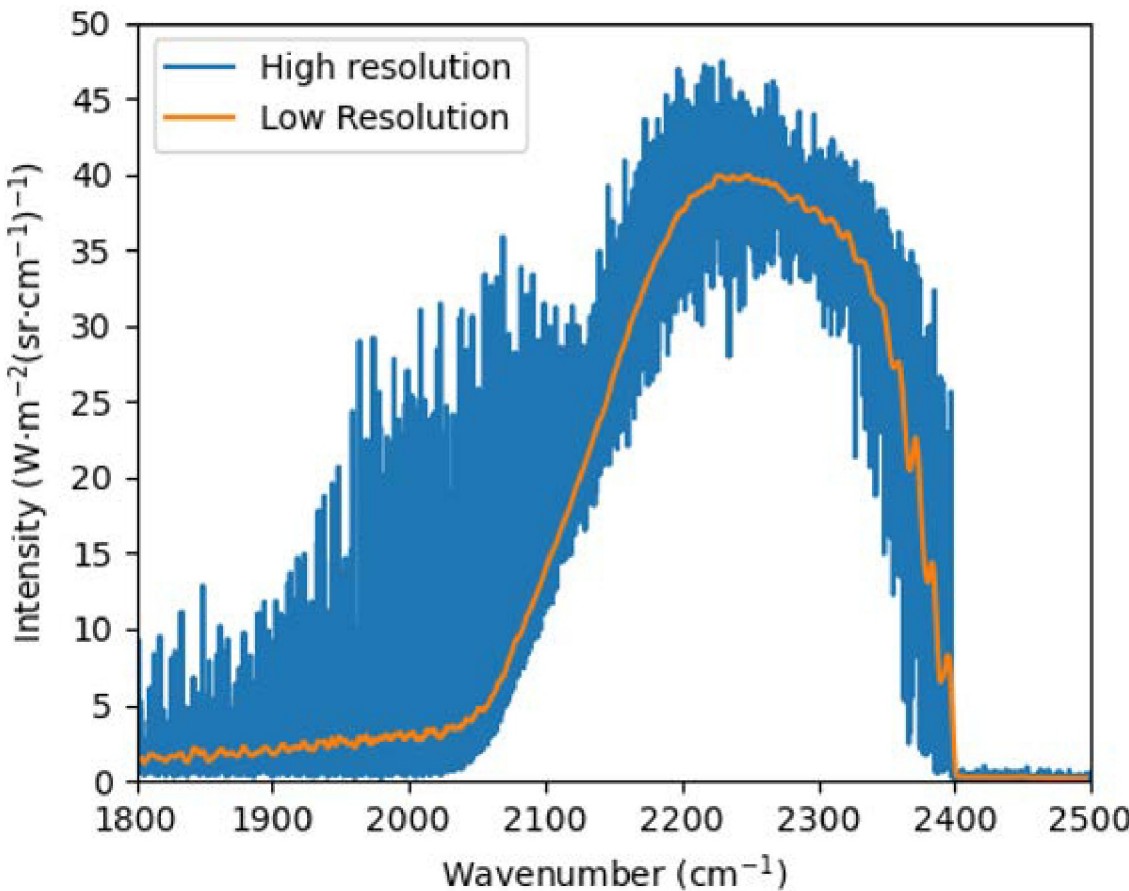

**Fig 4. A sample of low-resolution spectrum.**

and environmental monitoring settings. The temperature values in the dataset range from 1200 K to 3400 K, covering diverse temperature scenarios.

3. Spectral Resolution and Variability: The spectral data were generated with a low resolution of 4 cm$^{-1}$, subsequently blurred by instrument noise, to simulate the actual scenario, and also increase the difficulty in recovering temperature measurements. The low-resolution spectra were resampled every 0.1 cm$^{-1}$, resulting in 6,799 intensity values per spectrum.

## 3.2 Data-driven modeling

In the context of solving the inverse problem of line-of-sight emission spectroscopy, the temperature parameters along the light path were estimated from the generated spectral dataset via data-driven modeling. Two types of data-driven modeling methods were considered: (i) feature engineering and classical machine learning algorithms; and (ii) end-to-end convolutional neural networks (CNN). The methods explored in this research are shown in Fig 5. According to the performance comparison, the top performing methods are highlighted in Fig 5. The optimal system design consists of a combination of a series of feature engineering (including logarithmic transformation, signal representation feature extraction, PCA, and wrapper-based feature selection), and a two-stage machine learning method (weak learner + blender).

**3.2.1 Feature engineering.** Despite the substantial size of the generated spectra samples, the high dimensionality of the spectra, i.e., 6799 emission lines, would translate to a sparse sampling of the problem space, thus posing a challenge for ML algorithms. Therefore, feature engineering was performed on the raw spectra inputs, so as to extract informative features, thus reducing redundancy and problem dimensionality. First, the logarithmic transformation was usedcto reduce the degree of nonlinearity of the mapping between spectra and temperature. Next, two types of feature extraction methods were attempted, i.e., statistical and signal representation features, respectively. Principal Component Analysis (PCA) [30] was subsequently used to remove noise and irrelevant information, and further reduce the dimensionality of features. In the following subsections, we briefly introduce the feature engineering operations.

*3.2.1.1 Logarithmicic transformation.* According to Eq 4 in the physical forward model, the absorption coefficient, which contains the temperature information is affected by an exponential function. This introduces nonlinearities in the associated temperature-spectrum mapping, which need to be resolved through machine learning. The application of the logarithmic transformation [31] to the raw signals assists in reducing the degree of the mapping nonlinearity.

*3.2.1.2 Statistical features.* Statistical features, such as first-, second- and higher-order statistics are often used to describe high-dimensional data [32, 33]. The list of time domain statistical parameters estimated from the emission spectra is shown in Table 2. Moreover, the Fourier transform (FT) was applied on the spectra, and the extracted frequency domain features are shown in Table 3. In total, 38 features were obtained, 23 from the time domain, see Table 2, with the remaining 15 coming from the frequency domain, see Table 3. In addition to extracting these features by considering the entire waveband (1800-2500 $cm^{-1}$), it is also possible to consider extracting these features from the characteristic bands of the chemical species. By characteristic band, we refer to the part of the spectral range, where the signal of one mixture component is dominant, with negligible contributions originating from the other components. For H$_2$O, CO and CO$_2$, such characteristic bands are the ranges of (1800 $cm^{-1}$, 1890 $cm^{-1}$), (2100 $cm^{-1}$, 2190 $cm^{-1}$), and (2310 $cm^{-1}$, 2400 $cm^{-1}$), respectively. Thus, three further sets of 38 statistical features were also extracted from each of the three characteristic bands.

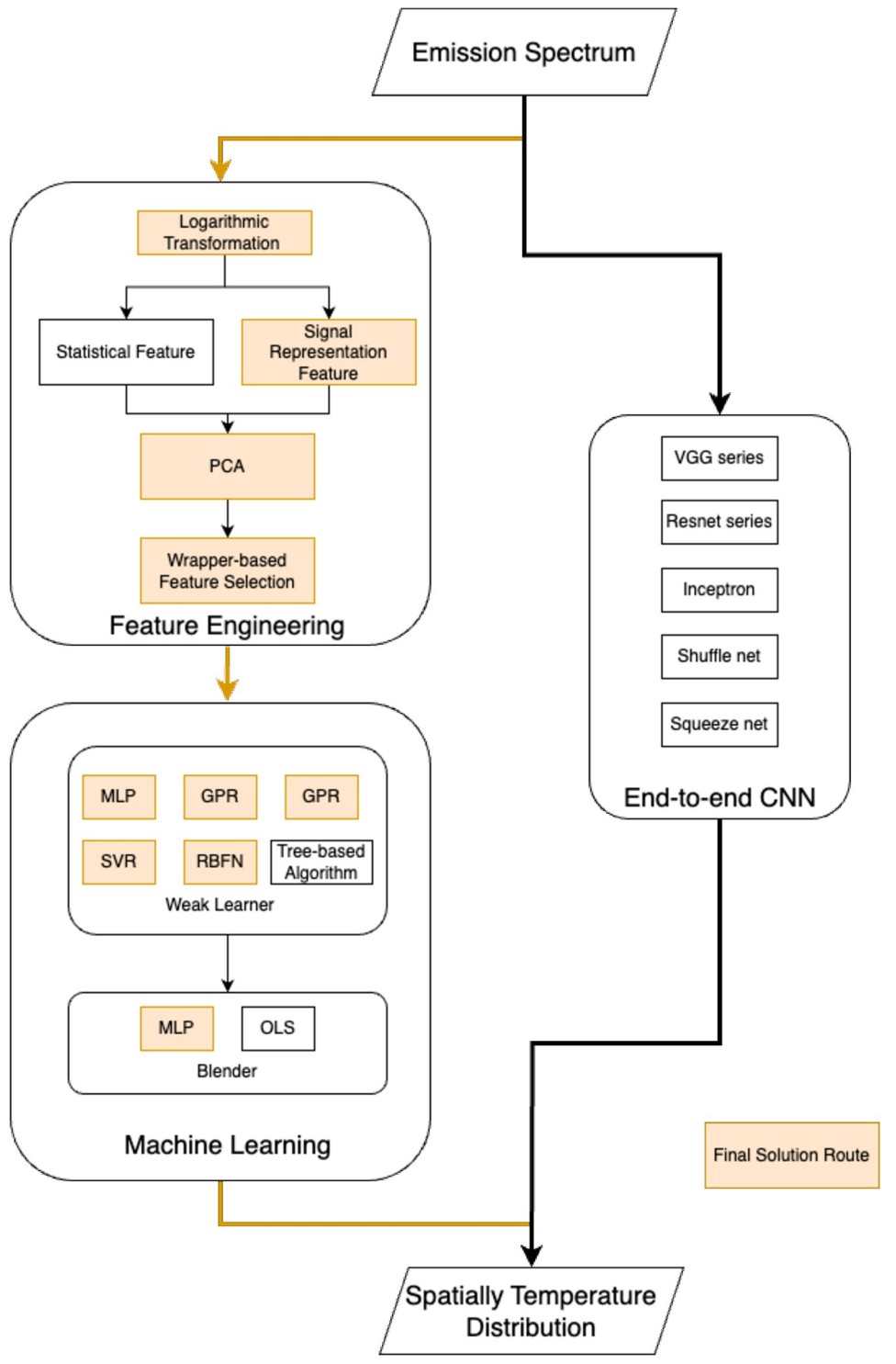

**Fig 5. Flowchart of the research methodology.** The optimal design combination is highlighted in boldface.

**Table 2. Time domain features of the emission spectra.**

| Features | Definition |
|---|---|
| Mean | $\bar{I} = \frac{1}{n}\sum_{i=1}^{n} I_i$, $i$ is the index of spectral signal, $n = 6799$ |
| Maximum | $I_{max} = max(I_i)$ |
| Minimum | $I_{min} = min(I_i)$ |
| Quartile 1 | $I_{0.25*n}$, $I$ in ascending order |
| Quartile 2 (median) | $I_{0.5*n}$, $I$ in ascending order |
| Quartile 3 | $I_{0.75*n}$, $I$ in ascending order |
| Interquartile range | $I_{0.75*n} - I_{0.25*n}$ |
| Standard deviation | $S = \sqrt{\frac{1}{n}\sum_{i=1}^{n} (I_i - I)^2}$ |
| Variance | $S^2$ |
| Skewness | $\frac{1}{n}\sum_{i=1}^{n} \left(\frac{I_i - I}{S}\right)^3$ |
| Kurtosis | $\frac{1}{n}\sum_{i=1}^{n} \left(\frac{I_i - I}{S}\right)^4$ |
| Inverse coefficient of variation | $I/S$ |
| Peak to peak | $I_{max} - I_{min}$ |
| Zero crossing rate | Number of signal sign changes |
| Root mean square | $RMS_I = \sqrt{\frac{1}{n}\sum_{i=1}^{n} I_i^2}$ |
| Crest factor | $I_{max}/RMS_I$ |
| Root mean square of difference | $S = \sqrt{\frac{1}{n-1}\sum_{i=1}^{n-1} (I_{i+1} - I_i)^2}$ |
| Root mean square of difference reciprocal | $S = \sqrt{\frac{1}{n-1}\sum_{i=1}^{n-1} \frac{1}{(I_{i+1} - I_i)^2}}$ |
| Mean of magnitude | $\frac{1}{n}\sum_{i=1}^{n} |I_i|$ |
| Difference variance | $\frac{1}{n-1}\sum_{i=1}^{n-1} |I_{i+1} - I_i|$ |
| Sum of differences | $\sum_{i=1}^{n} (I_{i+1} - I_i)$ |
| Shannon entropy of spectrum | $-\sum_{i=1}^{n} I_i^2 \log I_i^2; 0 \log 0 = 0$ |
| Log energy entropy of spectrum | $\sum_{i=1}^{n} \log I_i^2;\ \log 0 = 0$ |

*3.2.1.3 Signal representation features.* In [34], the coefficients of polynomial basis functions were used to represent intensities in the patches of a tomographic image. A similar idea was used in this research, where the spectra was decomposed into windows, each approximated by the coefficients of a basis function. These coefficients are termed signal representation features,

**Table 3. Frequency domain features of the emission spectra.**

| Features | Definition |
|---|---|
| Average of FT intensity | $\bar{I} = \frac{1}{n}\sum_{i=1}^{n} I_{F,i}$, $I_F$ is the signal in the FT domain, $i$ is the signal index in FT domain, $n = 6799$ |
| Average of FT magnitude | $\frac{1}{n}\sum_{i=1}^{n} |I_{F,i}|$ |
| Average of FT power | $\frac{1}{n}\sum_{i=1}^{n} I_{F,i}^2$ |
| Maximal power | $I_{max} = max(I_{F,i}^2)$ |
| Minimal power | $I_{min} = min(I_{F,i}^2)$ |
| Shannon entropy of FT | $-\sum_{i=1}^{n} I_{F,i}^2 \log I_{F,i}^2; 0 \log 0 = 0$ |
| Log energy entropy of FT | $\sum_{i=1}^{n} \log I_{F,i}^2;\ \log 0 = 0$ |
| Maximal magnitudes | The first 6 maximum magnitudes of the FT |
| Minimal magnitude | $min(|I_{F,i}|)$ |
| Average phase | $\bar{I} = \frac{1}{n}\sum_{i=1}^{n} A_{F,i}$ |

and provide a direct description of the local shape of the spectrum, rather than an abstract representation, as provided by the statistical features.

Two window lengths ($M$) were explored, i.e., 20 and 50 emission lines. The reason behind the choice of relatively small window lengths is based on the assumption that signal representation approaches are able to better approximate the spectral signal pattern inside a small window. However, if too small a window length is used, there is no substantial savings in terms of spectrum representation, compared to the raw signals. A variety of basis functions were used to approximate the spectra, i.e., polynomial, sinusoidal, exponential and power. In the experiments, polynomials of up to 3$^{rd}$ order were considered, while only first-order basis functions were selected for the remaining approximations. The benefit of doing so is that the raw spectral information within each window can be represented by up to four coefficients, resulting in compact signal representation, thus helping to tackle the curse of dimensionality.

The polynomial basis function approximation is given by:

$$\hat{I}_{x,P} = \sum_{k=0}^{n} a_k x^k \tag{12}$$

where $\hat{I}_{x,P}$ is the fitted spectral intensity at index $x$, obtained through polynomial fitting, $a_k$ is the coefficient for the kth power of $x$, $x$ is the local sample index inside the window, which varies from 1 to $M$, depending on the window size (i.e., M = 20, and 50 samples), $n$ is the highest order of the polynomial approximation. In these experiments, $n = 1, 2, 3$.

Next, the first-order sinuisoidal approximation, described by the four coefficients, $c$, $q_1$, $q_2$ and $f$, is given by:

$$\hat{I}_{x,F} = c + q_1 cos(fx) + q_2 sin(fx) \tag{13}$$

where $\hat{I}_{x,F}$ is the fitted spectral intensity at index $x$, obtained through sinusoidal fitting, $c$ is the constant term, $q_1$ and $q_2$ are the magnitudes of the cosine and sine terms, respectively, and $f$ is the frequency of the cosine and sine terms.

Similarly, the first-order exponential approximation, described by the four coefficients, $m_1$, $m_2$, $b_1$, $b_2$, is given by:

$$\hat{I}_{x,E} = c + m_1 exp(b_1 x) + m_2 exp(b_2 x) \tag{14}$$

where $\hat{I}_{x,E}$ is the fitted spectral intensity at index $x$, obtained through exponential fitting.

Last, the first-order power approximation, described by the three coefficients, $z$, $j$ and $c$, is given by:

$$\hat{I}_{x,PW} = zx^j + c \tag{15}$$

where $\hat{I}_{x,PW}$ is the fitted spectral intensity at index $x$, obtained through power fitting, $z$ is the magnitude of the power term, $j$ is the power coefficient, and $c$ is the constant term.

*3.2.1.4 Principal Component Analysis.* PCA is a popular method for feature selection and dimensionality reduction. It represents the original feature space by a group of orthogonal bases, which correspond to the directions of highest variance of the features. Thus, the original features were decomposed and represented in the bases' direction with decreasing order of magnitude of variance. Consequently, the order of the PCA-transformed features is indicative of their importance and information content, where the features with very low orders can be regarded as noise or unimportant information.

**3.2.2 Machine learning/Deep learning.** To ensure a robust and comprehensive investigation of temperature profile estimation, a diverse suite of machine learning models was employed, each representing distinct learning paradigms with complementary strengths. The chosen methods include Multi-Layer Perceptron (MLP) [35] for capturing complex non-linear relationships, Gaussian Process Regression (GPR) for its probabilistic predictions and suitability in small datasets [36], Support Vector Regression (SVR) [37] for its efficiency in high-dimensional spaces, and Radial Basis Function Networks (RBFN) [38] for effective interpolation and localized function approximation. Additionally, ensemble methods [39], i.e., Random Forests [40] and Boosted Trees [41] were utilized for their ability to combine multiple weak trees, offering robustness and improved generalization. In addition to the above ML models, a blending method is also employed to provide superior performance based on the aforementioned models' performance through a two-stage training process, as shown in Fig 6. The first stage models are the weak learners, i.e., the ML models used to tackle the problem individually. The second stage of the process is the blending model or meta learner, which provides the final estimates, using the primary estimates from the weak learners. The meta learner can be a linear (e.g., weighted average) or a nonlinear model (e.g., MLP). This diverse model selection ensures that complementary perspectives and capabilities are leveraged, enabling a systematic evaluation of the predictive performance across different machine learning approaches.

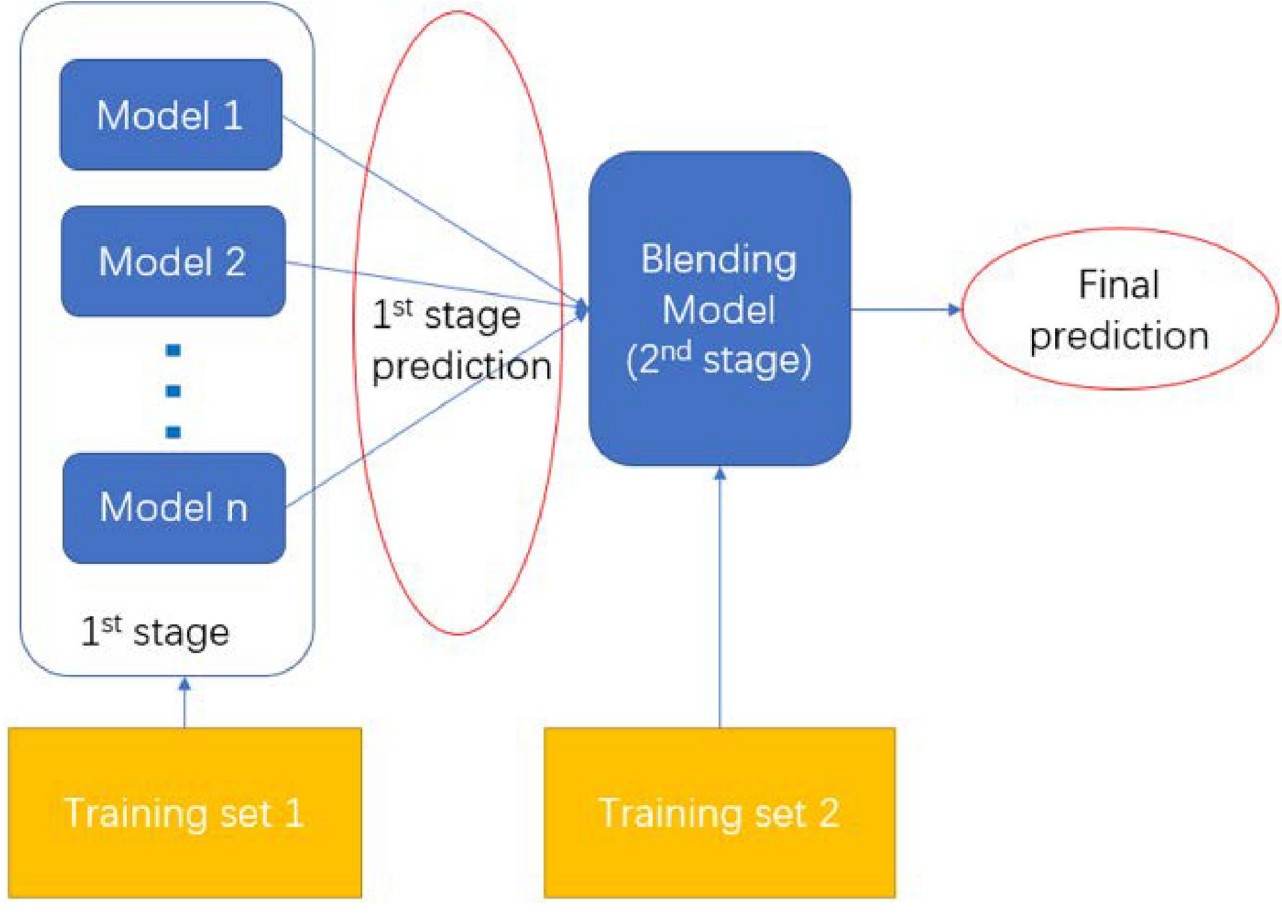

**Fig 6. Schematic of the blending method.**

In order to have a means of comparison for the combination approach of feature engineering and machine learning algorithms, the raw spectra was processed using state-of-the-art CNN architectures [42]. CNN is well-known for its ability to automatically extract features, thus alleviating the need for feature engineering. The CNN architectures investigated in this research are VGG series [43], Resnet series [44], Inception series [45, 46], Squeeze net [47] and Shuffle net [48]. The selection of VGG, Resnet, Inception, Shuffle Net, and Squeeze Net for comparison in this study was guided by their prominence in the field and their architectural diversity. VGG represents one of the earliest and most well-known convolutional neural network architectures, emphasizing deep stacking of layers for feature extraction. Resnet introduces the concept of residual learning through skip connections, which addresses the vanishing gradient problem in deep networks. Inception networks leverage multi-scale convolutional kernels, allowing them to extract features across different receptive fields, making them effective for varied data patterns. Shuffle Net and Squeeze Net, on the other hand, are lightweight architectures designed for computational efficiency, making them practical for scenarios with limited resources. These models were chosen as they collectively cover a wide spectrum of design philosophies, from classical architectures to modern lightweight designs, ensuring a comprehensive evaluation.

As these CNN models were originally designed for image classification tasks, some modifications were made prior to their use in the regression task of measuring spatially resolved temperatures. The main changes were as follows: (1) batch normalization and dropout were removed from the networks, since batch normalization shifts the center and standard deviation of the normalization operation with respect to the varied sampling of data, thus leading to increased bias and variance for the regression task, and (2) the last 399 spectral intensities were discarded, and the remaining 6400 intensities were reshaped to a square image of 80*80. The architectures utilized and modified for the purposes of the experiments are available from the open-source repository, https://github.com/RalphKang/Spatially_temp_measurement.

# 4 Results and discussion

The data-driven methodologies introduced in Sec. 3.2 were used in the experimental studies, and the corresponding results are provided below. Sections 4.1 and 4.2 report the performance of the combination of feature engineering and machine learning, and Convolutional Neural Networks, respectively. Sec. 4.3 provides a performance comparison between the top performing model in our experiments and the state-of-the-art approaches, demonstrating the effectiveness of the proposed approach. Sources of potential errors and a critical analysis of the limitations of the proposed method are discussed in Sec. 4.4.

## 4.1 Application of feature engineering and machine learning

The high dimensionality and sparsity of the raw spectral data may lead to divergence of ML models, which necessitates the use of feature engineering to extract and select valuable features for information representation. Motivated by the above observation, Sec. 4.1.1 demonstrates and discusses the performance of the proposed systematic feature engineering methodology, followed by the presentation and discussion of the optimal feature groups for machine learning modeling, and their corresponding performances in Sec. 4.1.2.

**4.1.1 Feature engineering.** First, the results of the application of feature engineering and machine learning methods are presented. The processing sequence followed for feature engineering was as follows: (i) physics-guided transformation, (ii) statistical/representation-based feature extraction, and (iii) PCA.

Following the logarithmic transformation, 38 statistical features were extracted from the entire spectrum, which were used as the first group of features. Moreover, we assumed that the information contained in the entire waveband and the characteristic wavebands of each of the specific chemical species were complementary to each other. Accordingly, a comprehensive description of the entire spectrum could be provided. Therefore, three additional feature groups, each containing 38 features, respectively, were extracted from the characteristic bands of $H_2O$, CO, and $CO_2$. The features extracted from the entire waveband and the specific wavebands were merged and used as an integrated feature group.

In the case of representation-based features, each spectrum signal was divided into windows of either 20 or 50 spectral samples, followed by their approximations using the four sets of basis functions. The approximation coefficients were then used to represent the information of the spectrum. By doing so, twelve groups of signal representation features were obtained, as shown in Table 4. The notation used for the reporting of the signal representation features involves their order and window length, e.g., $3^{rd}$-20 relates to third-order approximation being used to fit the window data, containing 20 samples.

In Table 4, the approximation quality of the various basis functions was compared by means of the Mean Square Error (MSE) and Pearson's correlation coefficient (R) between the original physics-transformed spectra and their reconstructions using signal representation features. In general, the majority of methods offer high quality approximation, as demonstrated in Table 4 and Fig 7, apart for the exponential, and sinusoidal basis functions with a window length of 20 samples. This is because the exponential basis functions are unable to accurately reconstruct the spectra as some of the coefficients become excessively large, leading to the overflowing of the exponential function. The reason behind this is the adversarial work style of the two exponential functions in the approximation formula, which could lead to the unbounded increase of the associated function coefficients.

In the case of the sinusoidal basis functions with a window length of 20, the values of $q_1$ and $q_2$ become excessively large in the case of smaller windows, thus leading to distortion. This is evident when comparing Fig 7(b), where a spike is observed, in the case of a window with length of 20 samples, against Fig 7(c), when a window of 50 samples is used. Apart from these two exceptions, the observation is that the higher the order of the fit, and the smaller the window length, the higher the quality of approximation. Moreover, the polynomial basis function approximation is the most accurate among all basis functions, for the same number of approximation coefficients, e.g., $2^{nd}$ order polynomials vs $1^{st}$ order power, and $3^{rd}$ order polynomials

**Table 4. Performance comparison of signal representation features.**

| Fitting basis function | | Feature number | MSE | R |
|---|---|---|---|---|
| Polynomial fitting | $3^{rd}$-20 | 1356 | **1.00E-05** | **1** |
| | $3^{rd}$-50 | 540 | 9.20E-05 | 0.9999 |
| | $2^{nd}$-20 | 1107 | 3.80E-05 | **1** |
| | $2^{nd}$-50 | 405 | 0.0003 | 0.9997 |
| | $1^{st}$-20 | 678 | 0.0001 | 0.9999 |
| | $1^{st}$-50 | 270 | 0.0011 | 0.999 |
| Sinusoidal fitting | 20 | 1356 | 0.3745 | 0.7623 |
| | 50 | 540 | 0.0002 | 0.9997 |
| Exponential fitting | 20 | 1356 | - | - |
| | 50 | 540 | - | - |
| Power fitting | 20 | 1017 | 9.50E-05 | 0.9999 |
| | 50 | 405 | 0.0005 | 0.9995 |

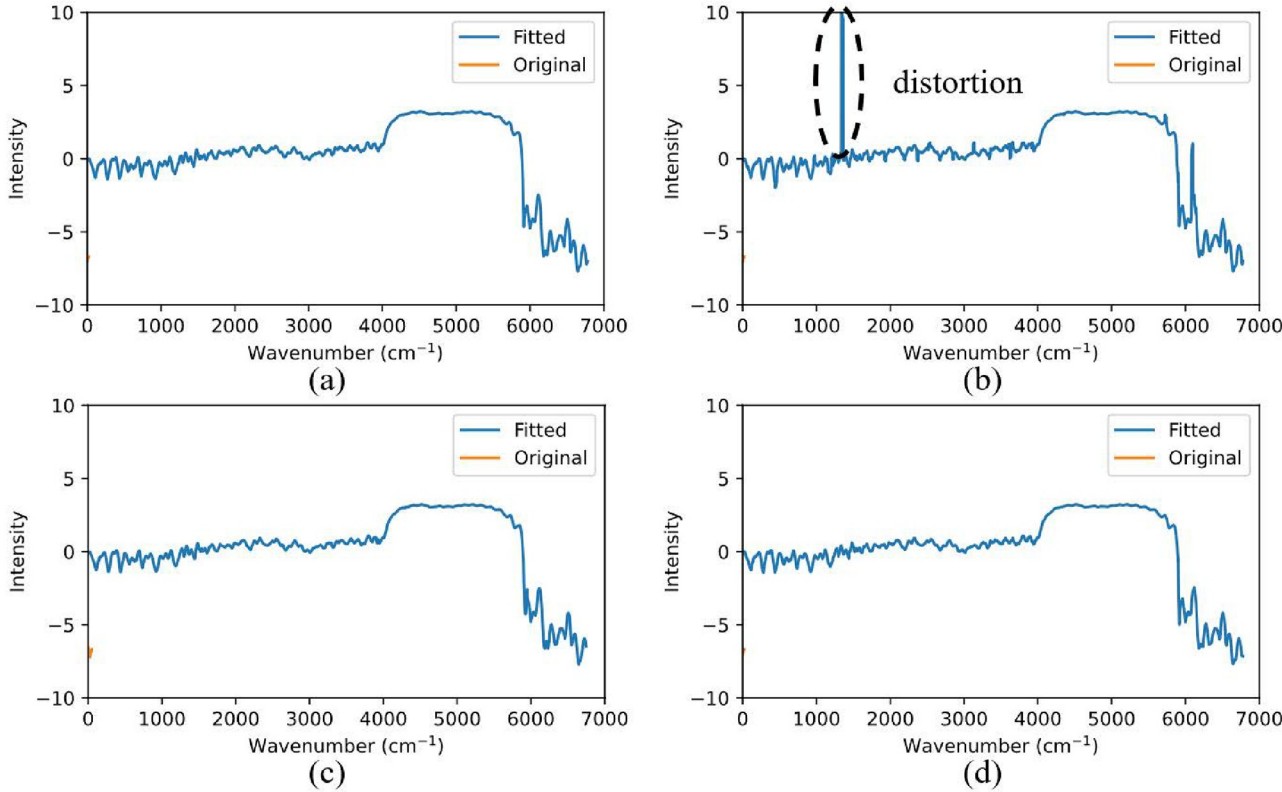

**Fig 7.** An example of spectrum approximation using: (a) $3^{rd}$ order polynomials and a window length of 20; (b) $1^{st}$ order sinusoidal basis functions and a window length of 20; (c) $1^{st}$ order sinusoidal basis functions and a window length of 50; (d) $1^{st}$ order power basis functions and a window length of 20.

vs $1^{st}$ order sinusoidal approximation. When comparing between polynomial approximations, the case of $3^{rd}$-50 has comparable accuracy to the case $2^{nd}$-20, while the former only has about half the number of features of the latter, and is much better than the case of $1^{st}$-20, although the latter has a larger number of features.

Following feature extraction, PCA was applied to all feature groups. Multiple variant feature groups were derived from the original feature groups by gradually eliminating the least informative PCA-transformed features. For example, using third-order polynomials and a window size of 50 resulted to 540 features, as the 6799 spectral inputs were divided into 135 windows, each window represented by four features. Following the application of PCA, all 540 PCA-transformed features were considered as a variant feature group, and subsequently reduced in order of decreasing variance to multiples of one hundred, i.e., 500,400,300, etc.

Finally, a wrapper-based feature selection method [49] was used to select the optimal feature groups, where an MLP was used as the referee, and the performance of the feature groups was assessed by its ability to approximate the spatial temperature profiles. The temperature values were normalized in the range of [0, 1] using the boundary values in each segment. In principle, this operation can improve the quality of learning. The spectra dataset was divided into training, validation, and test datasets with ratios of 70%, 15%, and 15%, respectively. The optimal performances achieved by the various feature groups are summarized in Table 5. The table provides information regarding the size of the original feature vector, the optimal feature vector size following the application of PCA, the optimal number of neurons in the hidden

**Table 5. Performance comparison of feature extraction methods.**

| Feature Category | Detailed feature description | | Raw Features | PCA Features | Hidden Neurons | Training MSE | Validation MSE |
|---|---|---|---|---|---|---|---|
| Statistical Features | Entire band | | 38 | 38 | 55 | 0.042 | 0.043 |
| | Characteristic bands | | 114 | 100 | 40 | 0.027 | 0.028 |
| | Feature aggregation | | 152 | 140 | 50 | 0.025 | 0.026 |
| Signal representation features | Polynomial basis | $3^{rd}$-20 | 1356 | 800 | 15 | 0.011 | **0.013** |
| | | $3^{rd}$-50 | 540 | 500 | 20 | 0.011 | **0.013** |
| | | $2^{nd}$-20 | 1107 | 700 | 20 | **0.010** | **0.013** |
| | | $2^{nd}$-50 | 405 | 405 | 20 | 0.012 | 0.014 |
| | | $1^{st}$-20 | 678 | 678 | 20 | **0.010** | **0.013** |
| | | $1^{st}$-50 | 270 | 200 | 25 | 0.015 | 0.016 |
| | Sinusoidal basis | 20 | 1356 | 1356 | 70 | 0.014 | 0.017 |
| | | 50 | 540 | 540 | 25 | 0.017 | 0.019 |
| | Exponential basis | 20 | 1356 | 1356 | 25 | 0.013 | 0.016 |
| | | 50 | 540 | 540 | 30 | 0.017 | 0.019 |
| | Power basis | 20 | 1017 | 1017 | 25 | 0.015 | 0.016 |
| | | 50 | 405 | 405 | 25 | 0.019 | 0.020 |

layer of the MLP, and the MSE of training and validation sets on the normalized temperature profiles.

As shown in Table 5, the statistical features acquired from the characteristic bands perform much better than those acquired from the entire waveband. This is due to the features extracted from the characteristic bands providing more relevant information. Moreover, the aggregation of statistical features further improves performance, which confirms the hypothesis that the information contained in the two statistical feature groups is distinct and complementary. However, the performance of statistical features is substantially inferior to that of representation-based features. This could be partly due to the number of extracted features. However, a more reasonable explanation is that statistical features are not as informative as representation-based features in terms of capturing the local shape changes of the emission spectra. This is supported by the observation that the best validation performance acquired by feature aggregation is merely 0.026, while the validation performance acquired by the lowest performing approximation, i.e., polynomial fitting of $1^{st}$-50 is 0.016, which offers an improvement of approximately 38%, with a modest increase of only 60 additional features.

Accordingly, the focus of the analysis shifts to signal representation features, with polynomial basis approximation providing a lower MSE. For instance, the lowest performing polynomial fit model, i.e., $1^{st}$ order with a window of 50 inputs, is comparable in terms of MSE with the top performing sinusoidal model, $1^{st}$ order with 20 samples, however, the former utilizes about one-sixth of the number of features of the latter. Indeed, the use of higher-order models and smaller window lengths leads to better performance, since they tend to provide a fine-grained approximation of the spectrum. However, this boosting effect tends to saturate, as third-order polynomial approximation with a window size of 50 does not demonstrate a performance improvement, compared to that with a window size of 20. The reason behind this is that the information contained in the spectrum can be appropriately represented by a small number of features, or equivalently, extracting a larger number of features through increased sub-division of the spectrum is redundant. Such a conclusion may be supported by the observation that the number of selected features following the application of PCA drops more significantly for smaller window sizes. Thus, third-order polynomial fitting with a window size of

50 samples offers a good compromise between feature complexity and approximation performance.

**4.1.2 Machine learning model performance.** Following the feature performance analysis of the previous section, the group of features obtained through the application of third-order polynomial approximation with a window length of 50 samples was used to train a variety of machine learning models. To support efficient convergence, all features were normalized in the range of [-1,1], with the outputs (temperatures of segments) normalized in the range of [0, 1]. Grid search was performed on the number of features, by varying the amount of features in multiples of one hundred. The temperature values were normalized in the range of [0, 1] by using the boundary values in each segment. The performance of the models is summarized in Table 6. In addition to the MSE on the normalized temperature values, the evaluation of the proposed models was conducted using four additional key performance metrics, i.e., Root Mean Square Error (RMSE), Relative Error (RE), Relative Root Mean Square Error (RRMSE), and the correlation coefficient (R). Each metric provides unique insights into the accuracy and reliability of the temperature estimations from emission spectra. Below is a detailed analysis of these metrics and their significance:

- Root Mean Square Error (RMSE): This measures the square root of the average squared differences between the predicted and actual temperature values. RMSE provides a clear indication of the model's overall predictive accuracy. A lower RMSE indicates that the model's predictions are closer to the actual temperature values, which is crucial for applications requiring precise temperature control. In practical temperature measurement scenarios, such as combustion monitoring and industrial process control, maintaining a low RMSE is essential in avoiding operational inefficiencies or safety issues. For example, a high RMSE in a combustion process might result in suboptimal fuel usage or increased emissions, leading to higher costs and environmental impact.

- Relative Error (RE): The Relative Error quantifies the absolute difference between the predicted and actual temperatures, expressed as a fraction of the actual temperature value. This provides insight into the accuracy of predictions relative to the magnitude of the actual

**Table 6. Performance comparison of machine learning models.**

| Machine Learning Model | Subclass | Feature number | Normalized Temperature | | Original Temperature (test set) | | | |
|---|---|---|---|---|---|---|---|---|
| | | | Training MSE | Testing MSE | RMSE (K) | RE | RRMSE | R |
| MLP | - | 500 | 0.011 | 0.0129 | 68.3 | 0.018 | 0.027 | 0.993 |
| RBFN | - | 500 | 0.014 | 0.0160 | 75.9 | 0.020 | 0.030 | 0.992 |
| SVR | Linear kernel | 540 | 0.015 | 0.0161 | 76.2 | 0.021 | 0.030 | 0.992 |
| | Gaussian kernel | 540 | 0.005 | 0.0144 | 72.2 | 0.019 | 0.028 | 0.993 |
| | Polynomial kernel | 540 | 0.006 | 0.0148 | 73.1 | 0.020 | 0.029 | 0.992 |
| GPR | Exponential kernel | 300 | **2E-07** | 0.0165 | 77.2 | 0.021 | 0.030 | 0.991 |
| | Rational quadratic kernel | 500 | **2E-07** | 0.0159 | 75.9 | 0.021 | 0.030 | 0.992 |
| | Matern 32 kernel | 200 | 4E-05 | 0.0193 | 83.4 | 0.023 | 0.033 | 0.990 |
| | Matern 52 kernel | 200 | 4E-07 | 0.0193 | 83.5 | 0.024 | 0.033 | 0.990 |
| | Square exponential kernel | 300 | 9E-07 | 0.0224 | 89.9 | 0.027 | 0.035 | 0.988 |
| Tree-based model | LSBoost | 540 | 0.022 | 0.0312 | 106.1 | 0.031 | 0.042 | 0.984 |
| | Random forest | 540 | 0.014 | 0.0397 | 119.6 | 0.037 | 0.047 | 0.979 |
| Blending model | Linear Blender | - | 0.012 | 0.0119 | 65.6 | **0.017** | 0.026 | **0.994** |
| | Heavy Blender | - | 0.012 | 0.0116 | 64.7 | **0.017** | **0.025** | **0.994** |
| | Light Blender | - | 0.012 | **0.0115** | **64.3** | **0.017** | **0.025** | **0.994** |

values. It is particularly useful when comparing the model's performance across different temperature ranges. A low RE value suggests that the model's predictions are consistently accurate across varying temperature levels. This is advantageous in applications such as environmental monitoring, where temperature variations may span a wide range, and consistent accuracy is required to make reliable assessments.

- Relative Root Mean Square Error (RRMSE): RRMSE is a normalized version of RMSE, which expresses the error as a fraction of the average of the actual values and is useful when comparing the model's performance across different datasets, and scenarios with varying temperature scales. A low RRMSE value indicates that the model performs well relative to the typical values of the temperature data. This is beneficial for assessing the model's robustness across different experimental conditions and when generalizing to new datasets that may have different temperature profiles.

- Correlation Coefficient (R): The correlation coefficient measures the strength and direction of the linear relationship between the predicted and actual temperatures. An R value close to 1 indicates a strong positive linear relationship, meaning that the model effectively captures the overall trend of temperature variations. A high R value is crucial for applications where understanding temperature trends and patterns is more important than absolute accuracy. For example, in climate modeling and atmospheric studies, capturing the general trend of temperature changes can be more informative for decision-making than minimizing individual prediction errors.

The combination of these metrics provides a comprehensive evaluation of model performance. For instance, a low RMSE and RE indicate high accuracy, making the model suitable for precise temperature measurements in industrial processes. A low RRMSE shows that the model's performance is consistent relative to the overall magnitude of temperature values, which is advantageous in varying experimental conditions. Finally, a high R value confirms that the model effectively captures temperature trends, making it valuable for applications such as environmental assessment and research in temperature dynamics.

Training was performed in two phases. In the first phase, weak learner models, e.g., MLP, RBF, SVR, GPR and tree-based models, were trained. The number of hidden neurons in the MLP and RBFN was optimized using grid search. The MLP network was configured to have a single hidden layer, and the number of hidden neurons was optimized in steps of 20, using grid search, and similarly, for RBFN. As for GPR, SVR, Random Forest, and LSBoost, these were optimized via Bayes optimization. Early stopping was applied to prevent overfitting. The models were trained for 5 times, to decrease the impact of variance on model assessment. The optimal checkpoint on the validation set was assessed on the test set. In the second phase of the training process, the blending method was used to further aggregate and enhance the performance of the base models.

The performance of the models in the first phase is shown in Table 6, with the MLP being the top performer. The intuition for this result is that since the MLP is used as a wrapper in feature selection, thus, the selected features are likely better suited for use by the MLP. It is observed that the remaining models exhibit satisfactory performance, which supports the choice of this feature group. However, this set of features cannot satisfy the requirements of all ML algorithms, as demonstrated by the performance of the tree-based methods.

In order to further improve prediction performance, the use of model blending was explored. Three variations of the blending model were attempted, namely, linear, heavy, and light blender. Their main differences are as follows: (1) Composition of weak learners: Linear

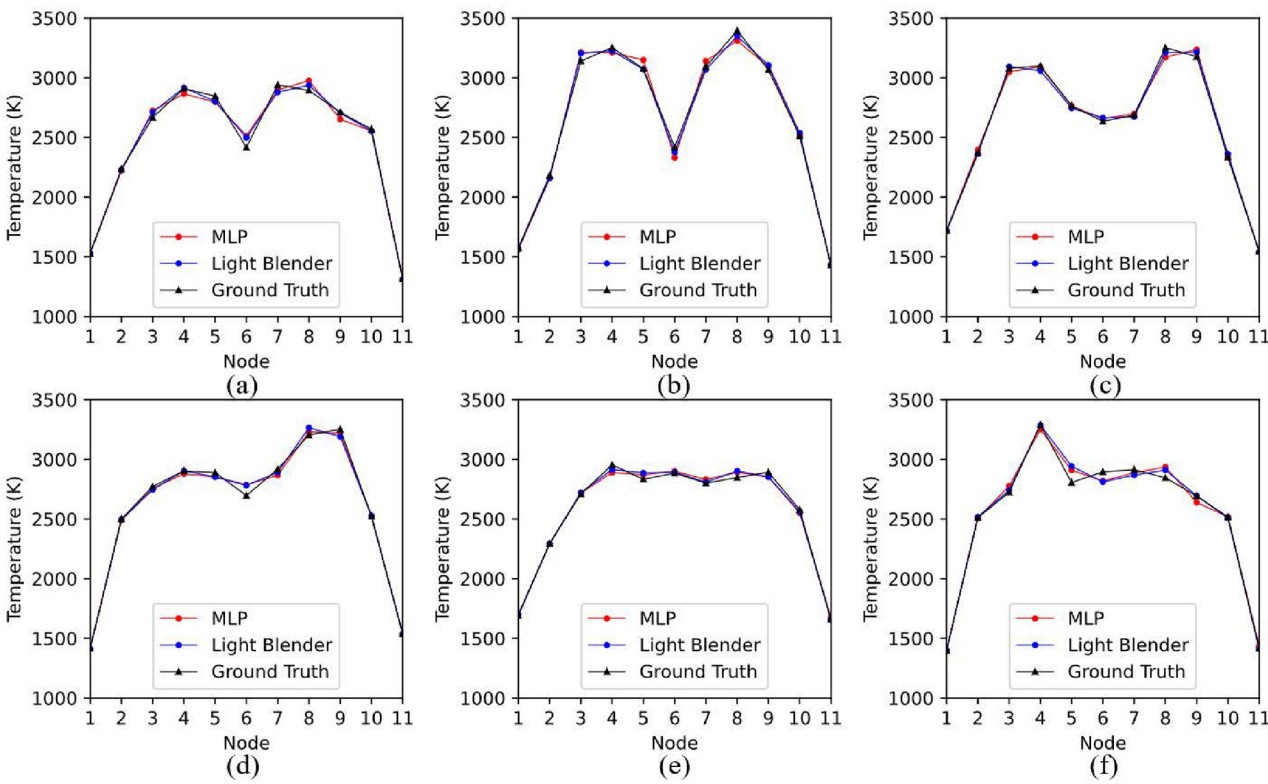

**Fig 8. Performance of MLP and light blender models in recovering non-uniform temperature profiles.** (a) Dual-peak Gaussian profile with a relatively small temperature variation; (b) Dual-peak Gaussian profile with a relatively large temperature variation; (c) Irregular dual-peak Gaussian profile with two "flatlands"; (d) Temperature profile with an increasing trend; (e) Trapezoidal temperature profile; (f) Temperature profile with a rush peak.

blender and heavy blender use all of the single-stage models as weak learners, whereas the light blender model drops the tree-based models because of their lower testing performance, and (2) Selection of meta learner: Linear blender utilizes ordinary Least Squares (OLS) estimation, while both heavy and light blenders employ the MLP architecture.

As shown in Table 6, the use of blending enhances reconstruction performance, as all three blending models perform better than the simple MLP model. However, it is also noted that this comes at the expense of increased computational and storage costs, as the blending models are based on multiple weak learners, including the MLP model. In addition, the MLP provides better nonlinear computation capabilities, and serves as a better meta learner than OLS. The inclusion of tree-based models as weak learners negatively impacts on the overall performance of the blending models, due to their weak base model performance.

Fig 8 displays representative temperature profile estimation examples on the test dataset, provided by the MLP and the Light Blender model. Both models work well, although their performance slightly degrades when dealing with gradually irregular profiles, as shown in Fig 8(e) and 8(f).

## 4.2 Convolutional neural networks

A popular alternative to the combination of feature engineering and machine learning is end-to-end deep learning, which is gradually becoming the dominant computational learning

paradigm. This family of methods offers the advantages of uniformity and ease of use, as the feature extraction process is automatically optimized and integrated within the learning process. Thus, the experimental evaluation of this work includes mainstream CNN architectures as alternative end-to-end solutions. A total of eleven CNN models, including VGG series, Resnet series, Inceptron series, Shuffle net and Squeeze net, were trained with the raw spectral measurements in the task of spatially resolved temperature reconstruction. Among them, VGG [43] is the most classical CNN architecture, which uses purely stacked convolutional layers, while ResNet [44] adds skip connections to achieve the concept of residual learning. Moreover, we evaluated Inception [45, 46], including its ultimate evolution Xception, which uses convolutional kernels of varying sizes to sense diverse receptive fields, Shuffle net [48] and Squeeze net [47], which are offer lightweight architectures with more modern designs.

During training, the Adam optimizer [50] was used, and a weight decay of $1e^{-4}$ was selected to alleviate overfitting. The learning rate was preliminarily determined by the method suggested in [51], however, it was further decreased as the selected deep learning models were prone to overfitting the training data, particularly, after removing the dropout and normalization operations, required for the regression application. Thus, the learning rate was varied and determined for each individual CNN architecture. In order to better navigate the training direction, the warm-up operation [52] was used to train the networks in the first ten epochs, during which, the learning rate was gradually increased from a tiny value to the selected learning rate. Such a soft activation of training supports the appropriate choice of training direction. During the normal training process, the cosine annealing learning rate schedule [52] was used, which helps to avoid falling into local minima.

The test performances of the CNN models are summarized in Table 7. Overall, all CNN models have similar performances, so that, to some extent, the results can represent the performance level, which can be reached by conventional CNN architectures with raw spectra as inputs. Strictly speaking, the best performance based on the metrics of Table 7 is achieved by Resnet18. However, when compared to the results obtained from the combination of feature engineering and classical machine learning methods, the temperature measurement estimates using CNN are worse than most of classical machine learning models. Specifically, when comparing Tables 6 and 7, it is apparent that the values of the RMSE, RE and RRMSE metrics for Resnet18 are almost double compared to those of the light blender model.

A more direct comparison between Resnet18 and the light blender model is provided in Fig 9. Although Resnet18 can satisfactorily capture the basic features of the morphology of

**Table 7. Performance comparison of CNN architectures.**

| CNN Model | Subclass | Normalized Temperature | | Original Temperature | | | |
|---|---|---|---|---|---|---|---|
| | | Train MSE | Test MSE | RMSE | RE | RRMSE | R |
| VGG | VGG11 | 0.032 | 0.037 | 114.3 | 0.033 | 0.045 | 0.981 |
| | VGG13 | 0.032 | 0.036 | 113.7 | 0.033 | 0.045 | 0.981 |
| | VGG16 | 0.033 | 0.038 | 116.9 | 0.034 | 0.046 | 0.980 |
| | VGG19 | 0.040 | 0.043 | 124.1 | 0.037 | 0.049 | 0.978 |
| Resnet | Resnet18 | 0.026 | **0.033** | **108.9** | **0.031** | **0.043** | **0.983** |
| | Resnet34 | 0.031 | 0.035 | 111.0 | 0.032 | 0.044 | 0.982 |
| | Resnet50 | 0.028 | 0.033 | 109.4 | 0.032 | **0.043** | 0.983 |
| Inception | V3 | 0.033 | 0.036 | 113.4 | 0.033 | 0.044 | 0.981 |
| | Xception | 0.032 | 0.035 | 112.9 | 0.033 | 0.044 | 0.982 |
| Shuffle net | - | 0.032 | 0.036 | 113.4 | 0.033 | 0.044 | 0.981 |
| Squeeze net | - | 0.034 | 0.039 | 119.2 | 0.035 | 0.047 | 0.979 |

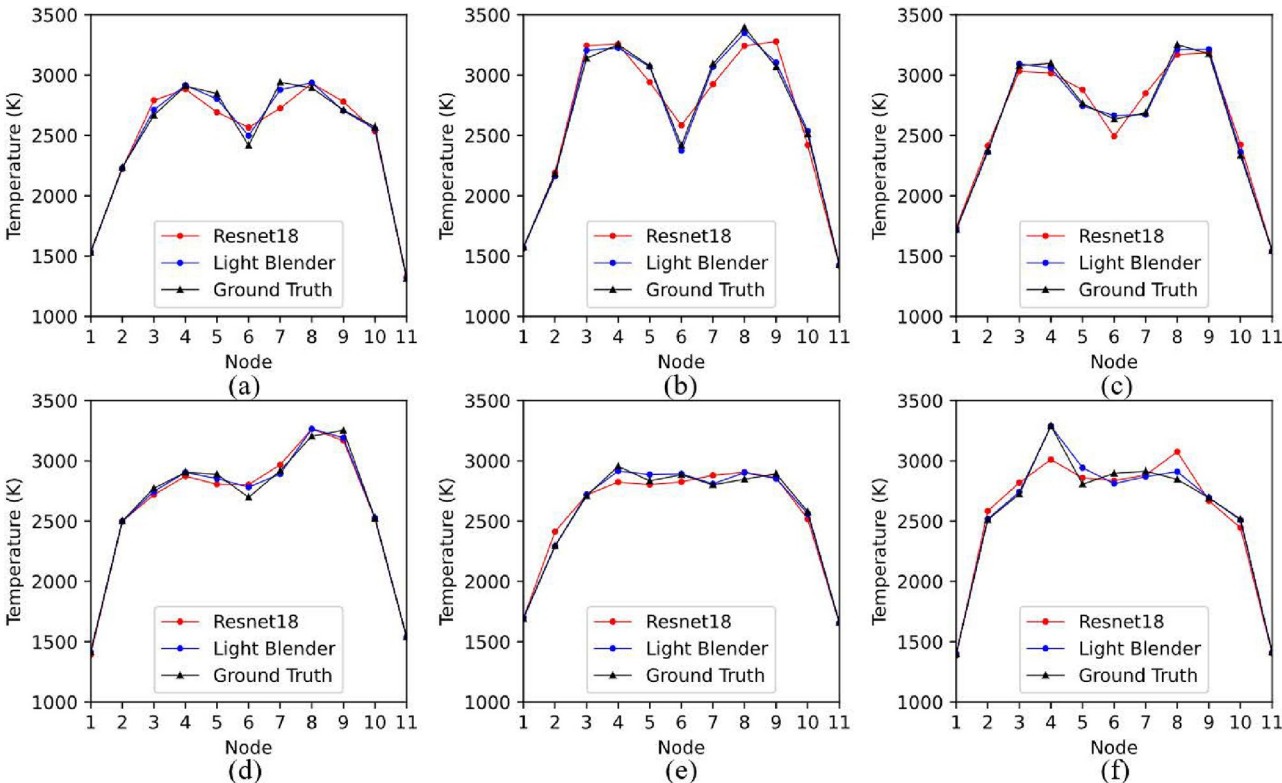

**Fig 9. Comparison between Resnet18 and the light blender models.** (a) Dual-peak Gaussian profile with a relatively small temperature variation;(b) Dual-peak Gaussian profile with a relatively large temperature variation; (c) Irregular dual-peak Gaussian profile with two "flatlands"; (d) Temperature profile with an increasing trend; (e) Trapezoidal temperature profile; (f) Temperature profile with a rush peak.

temperature profiles, it cannot provide as accurate temperature estimations as the light blender model. This comparison indicates that effective feature engineering, which is often underestimated in favor of the built-in feature extraction of CNN architectures in the state-of-the-art [53–55] can enable classical machine learning algorithms to achieve competitive performance.

Despite the well-documented success of Convolutional Neural Networks in various data-driven applications such as image classification, object detection, and time-series forecasting, their performance in the context of spatially resolved temperature measurement using emission spectroscopy was found to be inferior to the combination of advanced feature engineering and classical machine learning methods in this study. Several key reasons contribute to this outcome:

- Loss of Spectral Characteristics During Convolution: Spectral data is characterized by its peaks and troughs, which directly correlate with the physical properties of the gas mixture, such as concentration and temperature distribution. Convolutional filters, designed for translation invariance in images, do not inherently capture these spectral characteristics effectively. As a result, important spectral features such as peak intensities and their precise locations might be averaged out during convolution, leading to a loss of critical information necessary for accurate temperature estimation.

- Challenges with High-Dimensional Spectral Inputs: The high-dimensional nature of the spectral data (e.g., 6799 emission lines) poses a challenge for CNNs, which rely on down-sampling through pooling layers to reduce computational complexity. However, this

reduction can lead to information loss, especially when the input spectra contain fine-grained details that are crucial in distinguishing subtle variations in temperature. In contrast, feature engineering methods, e.g., polynomial approximations and PCA, preserve this high-dimensional information more effectively by focusing on extracting relevant features without losing the global context of the spectral data.

- Insufficient Feature Representation Learning: CNNs tend to learn hierarchical features through multiple layers of convolutional filters, which is highly beneficial for image data. However, in the case of emission spectra, where the relevant features are not hierarchical in nature, the learned representations may not align with the true underlying patterns. As demonstrated in this study, feature engineering techniques such as physics-guided transformations and signal representation features provide a more direct representation of the spectral data, allowing classical machine learning models to effectively leverage these features.

- Feature Engineering with Domain-Specific Knowledge: The feature engineering techniques employed in this study incorporate domain-specific knowledge through physics-based transformations and signal representation features. They are explicitly designed to capture the relationships between temperature, species concentration, and spectral intensities. This contrasts to CNNs, which lack such domain-specific priors and rely entirely on the data to learn these relationships. As a result, CNN methods often require larger datasets and extensive training to match the performance of models utilizing engineered features.

## 4.3 Comparison with the state-of-the-art

The methods described in [20, 21] were applied to the dataset of this study. To the best of the authors' knowledge, these are the only two studies where data-driven models are used to measure spatially resolved temperature profiles from emission spectroscopy. Both methods employ the MLP model. In [20], the conventional single hidden layer architecture is employed, whereas in [21], three hidden layers are used. As part of this comparison, the MLP architectures in these studies were precisely replicated. Grid search was performed to fine-tune the hyperparameters of the associated MLP models. In terms of feature engineering, no feature engineering is used in [21], rather, the raw spectra data is directly fed to the MLP. In [20], the spectrum is divided into windows of fixed length, and the ratios of line intensities at the two sides of the spectrum are used as features. Two window lengths of 10 and 20 samples were explored in this work.

In summary, three MLP models, two with the setup of [20], and one with the architecture of [21], were developed and trained. As demonstrated in Table 8, the light blender model (see Table 6) offers significant performance improvements, compared to these three models. A visual comparison of the performance of the two methods compared to the light blender model is given in Fig 10, where it can be observed that, in general, the methods of [20, 21] capture the properties of the dual-peak Gaussian profiles, which constitute the majority of the training set, but have very limited adaptability to irregular profiles, although the method in

**Table 8. Performance comparison of the state-of-the-art methods.**

| Method | Model | Input | RMSE | RE | RRMSE | R |
|---|---|---|---|---|---|---|
| Ren, et al. [21] | MLP (Three hidden layers) | Raw spectrum | 165.7 | 0.055 | 0.065 | 0.96 |
| cieszczyk, et al. [20] | MLP | Intensity ratios, window length of 50 samples | 130.5 | 0.04 | 0.051 | 0.975 |
| | MLP | Intensity ratios, window length of 20 samples | 130 | 0.04 | 0.051 | 0.976 |

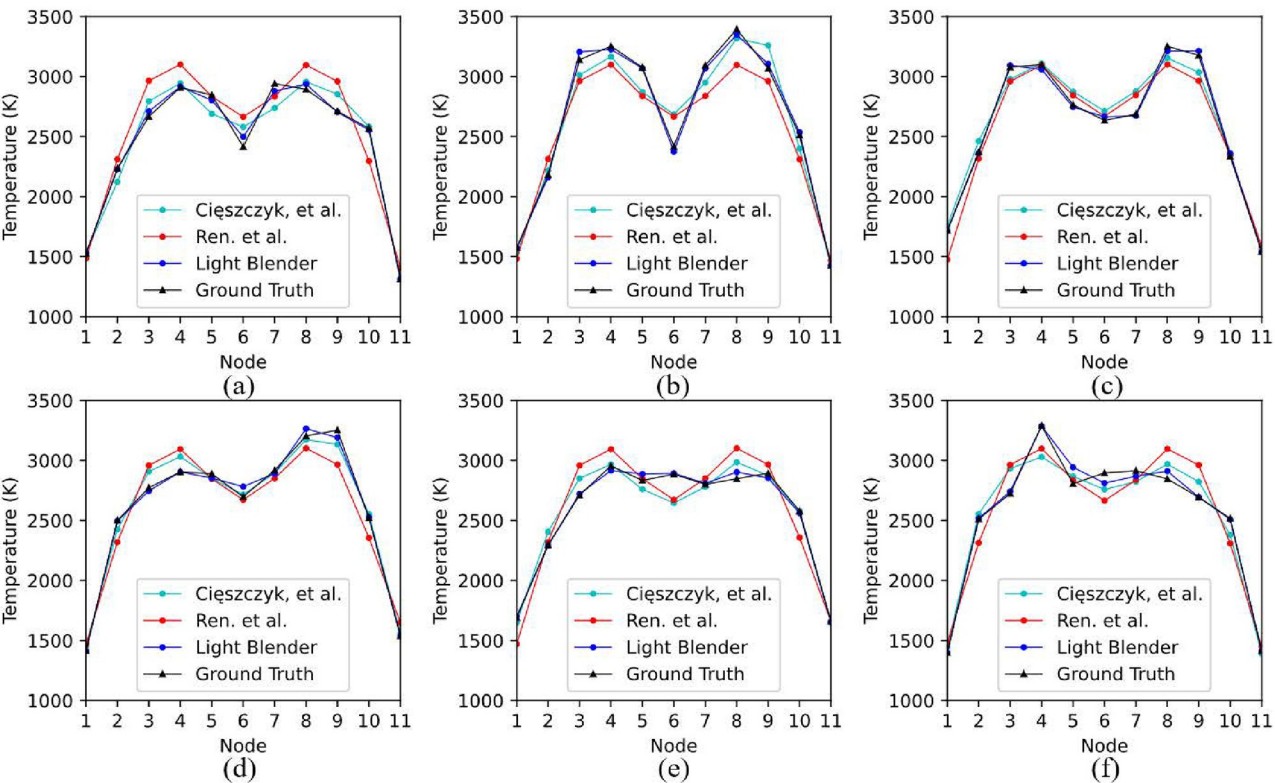

**Fig 10. Comparison between the state-of-the-art approaches of [20, 21] and the light blender model.** (a) Dual-peak Gaussian profile with a relatively small temperature variation;(b) Dual-peak Gaussian profile with a large temperature variation; (c) Irregular dual-peak Gaussian profile with two "flatlands"; (d) Temperature profile with an increasing trend; (e) Trapezoidal temperature profile; (f) Temperature profile with a rush peak.

[20] performs slightly better. According to the processing procedure, the main reason driving the significant performance gap is feature engineering. State-of-the-art methods pay little attention to the value of feature engineering, and accordingly, their ability to process high-dimensional spectra is challenged, affecting the robustness of their performance. In contrast, feature engineering, which has been systematically explored in this study, effectively captures the useful information of the spectra, whilst reducing the amount of noisy information, thus providing the ML models with the necessary foundation for robust reconstruction performance.

## 4.4 Error analysis

Although the models evaluated in this research work well, we performed an analysis of potential error sources, which could limit their performance in practice:

1. Model Uncertainty: Model uncertainty refers to the variability in predictions due to the inherent limitations of the model architecture, parameter choices, and the training process. This type of uncertainty is also known as aleatoric uncertainty, which arises from the stochastic nature of the learning algorithm, such as the use of random initial weights in neural networks or random splits in decision trees. In the context of emission spectroscopy, model uncertainty is often introduced by suboptimal model configurations, inappropriate model complexity (e.g., underfitting or overfitting), and the inability of the model to capture the complex non-linear relationships between spectral features and temperature distributions.

2. Data Uncertainty: Data uncertainty, also referred to as measurement or observational uncertainty, is caused by the inherent noise present in the emission spectra due to factors such as instrumentation limitations, fluctuations in the light source, and environmental conditions during data acquisition. The level of data uncertainty is directly influenced by the spectral resolution, the signal-to-noise ratio (SNR), and the quality of the measurements. In emission spectroscopy, noise may manifest itself as random variations in intensity values, baseline shifts, or the presence of spurious peaks that do not correspond to actual physical phenomena.

3. Epistemic Uncertainty: Epistemic uncertainty arises from the lack of knowledge or information about the underlying data distribution and the limitations in the training data. This type of uncertainty, also known as systematic or structural uncertainty, is present when the model encounters scenarios or data points that were not represented in the training set. In the context of temperature estimation from emission spectra, epistemic uncertainty may manifest as the model's inability to generalize to unseen temperature profiles, gas compositions, or spectral regions with unusual characteristics. It typically occurs when there is an insufficient number of training samples to cover the entire range of possible temperature distributions or when there are gaps in the data that prevent the model from learning the full data manifold. Epistemic uncertainty can lead to poor generalization and erroneous predictions when the model is exposed to out-of-distribution (OOD) data.

## 5 Conclusion and future work

In this study, the feasibility of using data-driven models to recover spatially resolved temperature distributions from line-of-sight emission spectroscopy data was systematically explored, focusing on two primary approaches, i.e., feature engineering with classical machine learning methods, and end-to-end convolutional neural networks. The research yielded several significant findings. First and foremost, the combination of physics-guided transformation, polynomial approximation-based features, and Principal Component Analysis emerged as the most effective feature extraction methodology. Next, among fifteen traditional machine learning models, the light blender model demonstrated exceptional performance, achieving remarkable metrics including a root mean square error of 64.3, relative error of 0.017, relative root mean square error of 0.025, and an impressive R-value of 0.994. This result not only validates the feasibility of data-driven models in measuring nonuniform spatial temperature distributions but also highlights the substantial advantages of feature engineering.

While end-to-end Convolutional Neural Networks offered competitive performance compared to state-of-the-art techniques, they could not match the performance of feature engineering and machine learning models. This finding underscores the critical role of carefully engineered features in enhancing predictive accuracy for spectroscopic temperature estimation.

The implications of this research extend beyond methodological advancements. The high-performing models of this research have significant potential for improving temperature estimation accuracy across various industrial and research domains. In industrial settings, such as combustion monitoring, materials processing, and chemical reactors, they can enable more precise temperature control, leading to improved operational efficiency, enhanced energy savings, and increased safety. The ability to accurately capture temperature variations across a region opens new possibilities for real-time monitoring systems, potentially reducing manual interventions and facilitating more automated, intelligent measurement techniques.

Looking forward, several promising avenues for future research have been identified. These include exploring the impact of incorporating new data to enhance the performance of both weak learners and meta learner, through retraining and fine-tuning. The objective of this investigation being to optimize the use of the blending approach to leverage larger data populations, potentially leading to improvements in measurement accuracy and reliability. Along a similar direction, we intend to increase the diversity of the datasets to consider broader application scenarios, systematically investigating the impact of hyper-parameter selection, considering advanced feature learning technologies, e.g., contrastive and self-supervised learning, and the use of advanced deep learning architectures and hybrid machine learning approaches, which combine feature engineering with deep learning architectures. Such studies could further refine and extend the capabilities of data-driven temperature estimation techniques, pushing the boundaries of spectroscopic measurement technologies.

## Acknowledgments

R.K. would like to acknowledge Dr Hongxia Li and Prof TJ Zhang at Khalifa University for their sincere help in sharing ideas in the initial stages of this research.

## Author Contributions

**Conceptualization:** Ruiyuan Kang, Dimitrios C. Kyritsis, Panos Liatsis.

**Data curation:** Ruiyuan Kang.

**Formal analysis:** Ruiyuan Kang.

**Investigation:** Ruiyuan Kang.

**Methodology:** Ruiyuan Kang, Panos Liatsis.

**Software:** Ruiyuan Kang.

**Supervision:** Dimitrios C. Kyritsis, Panos Liatsis.

**Validation:** Ruiyuan Kang.

**Visualization:** Ruiyuan Kang.

**Writing – original draft:** Ruiyuan Kang.

**Writing – review & editing:** Dimitrios C. Kyritsis, Panos Liatsis.

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
