## [Decision Letter · Decision Letter 0]

24 Sep 2024

PONE-D-24-35923A data-driven method for spatially resolved Thermometry from Line-of-Sight Emission SpectroscopyPLOS ONE

Dear Dr. Liatsis,

Thank you for submitting your manuscript to PLOS ONE. After careful consideration, we feel that it has merit but does not fully meet PLOS ONE’s publication criteria as it currently stands. Therefore, we invite you to submit a revised version of the manuscript that addresses the points raised during the review process.

We look forward to receiving your revised manuscript.

Kind regards,

Anurag Sinha, Ph.D

Academic Editor

PLOS ONE

Journal Requirements:

2. In the online submission form you indicate that your data is not available for proprietary reasons and have provided a contact point for accessing this data. Please note that your current contact point is a co-author on this manuscript. According to our Data Policy, the contact point must not be an author on the manuscript and must be an institutional contact, ideally not an individual. Please revise your data statement to a non-author institutional point of contact, such as a data access or ethics committee, and send this to us via return email. Please also include contact information for the third party organization, and please include the full citation of where the data can be found.

Reviewers' comments:

Reviewer's Responses to Questions

**Comments to the Author**

1. Is the manuscript technically sound, and do the data support the conclusions?

Reviewer #1: Partly

Reviewer #2: Yes

Reviewer #3: Yes

2. Has the statistical analysis been performed appropriately and rigorously? 

Reviewer #1: Yes

Reviewer #2: Yes

Reviewer #3: Yes

3. Have the authors made all data underlying the findings in their manuscript fully available?

Reviewer #1: Yes

Reviewer #2: Yes

Reviewer #3: Yes

4. Is the manuscript presented in an intelligible fashion and written in standard English?

Reviewer #1: Yes

Reviewer #2: Yes

Reviewer #3: Yes

5. Review Comments to the Author

Reviewer #1: 1. Some Keywords are not refined enough (e.g The human body posture estimation)

2. The introduction can be enriched by briefly describe the state-of-the-art in the title of this study and provide more recently related references to support foundation of this studies to better context for the current research. I will recommend citing recently published related papers;

a. Internet of Medical Things (IoMT): Applications, Challenges, and Prospects in a Data-Driven Technology. In: Chakraborty, C., Khosravi, M.R. (eds) Intelligent Healthcare. pp 299–319, Springer, Singapore. https://doi.org/10.1007/978-981-16-8150-9_14

b. Statistical Analysis of Stakeholders Perception on Adoption of AI/ML in Sustainable Agricultural Practices in Rural Development. In X. S. Yang, S. Sherratt, N. Dey, & A. (. Joshi (Ed.), Proceedings of Ninth International Congress on Information and Communication Technology. ICICT 2024 2024. Lecture Notes in Networks and Systems. 1003. Springer, Singapore. doi:https://doi.org/10.1007/978-981-97-3302-6_11

3. The problem that author is trying to address in this paper is not clear yet, (clearer motivation for study is required)

4. Add the contribution of this study and study organization towards end of Introduction Section

5. In 4-5- lines, authors should summarize the literature gaps identified before starting the methodology (Kindly relate your work in the context of existing work)

6. Author should re-check the correctness of equations especially equations 8 and 14

7. Services of language expert are required

8. Image quality needs to be improved

9. Ensure that all Figures are properly labeled and referenced

10. Authors should include the future work of this study

Reviewer #2: 1. The title is clear and concise. However, consider specifying in the title that you compare different data-driven models, as this may better capture the comprehensive scope of your research.

2. Your systematic exploration of data-driven models is well-structured. Ensure that the methodology section provides detailed explanations of the implementation of feature engineering techniques, including physics-guided transformations, polynomial approximation-based features, and PCA.

3. The methodology, results, and conclusions are clearly presented. Consider adding more discussion on practical implications and potential applications of your findings.

4. It is important that the description of the CNN models includes details on their architecture, training process, and evaluation metrics to provide a complete understanding of their performance relative to classical methods.

5. The presentation of performance metrics (RMSE: 64.3, RE: 0.017, RRMSE: 0.025, R: 0.994) is effective in highlighting the superiority of the light blender model. Ensure that the implications of these metrics are discussed in the context of practical applications

6. The acknowledgment that end-to-end CNNs are competitive but not superior is crucial. Consider providing more details on the specific limitations of CNNs in this context.

7. Expand on the architecture of the CNN models used in the study. Describe the layers, activation functions, and any preprocessing steps involved. Also, include details on the training process, such as hyperparameters and training duration.

8. Offer a more in-depth comparative analysis between feature engineering methods and end-to-end CNNs. Include a discussion on why feature engineering methods outperform CNNs in this context and any potential reasons for CNNs' lower performance.

9. Provide a detailed analysis of the performance metrics (RMSE, RE, RRMSE, and R) and explain their significance. Discuss how these metrics translate to practical advantages or limitations in temperature measurement.

10. Elaborate on the practical implications of the results. How do they impact the field of spatially resolved thermometry and emission spectroscopy? Discuss any potential applications or improvements in technology that could result from your findings.

11. Include an analysis of potential sources of error in your measurements and predictions. Discuss how these errors could affect the reliability of your method and any steps taken to mitigate them.

12. Provide a detailed description of the datasets used, including their size, diversity, and any preprocessing steps. This will help readers understand the context and limitations of the data.

13. I recommend conducting additional experiments to test your proposed method in varied conditions or with different types of emission spectra to further validate its robustness.

14. Your conclusion appropriately emphasizes the effectiveness of the feature engineering methodology. Ensure that the rationale for why these methods are superior is clearly articulated.

15. The conclusion effectively communicates the strong performance of the light blender model. It might be helpful to elaborate on how these results impact the field of spatially resolved thermometry.

Reviewer #3: In this paper, the authors have conducted a study on the enhancement of spatially resolved thermometry using line-of-sight emission spectroscopy by applying data-driven models to improve temperature measurement accuracy in non-homogeneous fields. While the study addresses a challenge in emission spectroscopy, several suggestions are provided to improve the clarity and depth of the methodology before recommending it for publication:

1. The authors should look into hybrid machine learning approaches, where feature engineering is used to augment deep learning models? This could potentially offer further improvements in prediction accuracy.

2. Please provide a more detailed explanation or justification of the assumption on thermal equilibrium and the use of dual-peak Gaussian functions for temperature profiles, particularly in scenarios where these conditions might not hold true, for example, turbulent flows.

3. The Related Works sections consist of very little information, therefore, I would recommend the authors to incorporate more information regarding previous research and their outcomes in this specific area of using data-driven models for temperature measurement. Additionally, adding a table as well to highlight the previous research done in this field and how it compares to this research.

4. I would suggest the authors to include a section focusing on uncertainty quantification and how the model's confidence intervals or error margins behave under various conditions

5. I would highly suggest that the section headings are presented in a numerical enumeration to keep a consistently organized flow for other researchers to follow.

6. I would recommend the authors to include additional benchmark algorithms or techniques other than just feature engineering and classical machine learning algorithms, and end-to-end convolutional neural networks to provide a more in-depth comparison on both qualitative and quantitative metrics

7. I would suggest the author to provide a more comprehensive explanation of why CNNs underperformed, especially given their success in many data-driven applications. This can add depth to the discussion.

8. I would suggest to add a section that explains why such particular architectures chosen for this regression task. For example, did any architecture outperform others in preliminary tests, or were the choices based on general trends in the field?

9. I would highly suggest the authors to add a comparison of model performance on low-resolution versus high-resolution spectra which could clarify the trade-offs involved in using different spectral resolutions

10. The research includes only two window lengths, 20 and 50 emission lines. I would suggest that the authors try to incorporate more than just these two window lengths.

6. PLOS authors have the option to publish the peer review history of their article (what does this mean?). If published, this will include your full peer review and any attached files.

Reviewer #1: **Yes: **Sunday Ajagbe

Reviewer #2: No

Reviewer #3: **Yes: **Shraiyash Pandey

---

## [Author Response · Author response to Decision Letter 0]

6 Nov 2024

Comments to the Author

We would like to sincerely thank all the reviewers for their insightful comments and constructive suggestions. Your thorough evaluation and thoughtful feedback have greatly contributed to improving the clarity, depth, and overall quality of the submitted manuscript. We truly appreciate the time and effort you have dedicated in reviewing our work, and your valuable input has been instrumental in refining our research. Thank you once again for your contributions to this process.

Reviewer #1: 

1.1 Some Keywords are not refined enough (e.g The human body posture estimation)

Response: Thank you for your valuable comment. We appreciate your suggestion regarding refining the keywords. We propose using "Spatially Resolved Thermometry, Feature Engineering, and Machine Learning" as more precise keywords to better reflect the focus of our study. However, we would like to note that PLOS ONE does not provide an option to set keywords directly. We hope this clarification helps and thank you again for your insight.

1.2 The introduction can be enriched by briefly describe the state-of-the-art in the title of this study and provide more recently related references to support foundation of this studies to better context for the current research. I will recommend citing recently published related papers;

a. Internet of Medical Things (IoMT): Applications, Challenges, and Prospects in a Data-Driven Technology. In: Chakraborty, C., Khosravi, M.R. (eds) Intelligent Healthcare. pp 299–319, Springer, Singapore. https://doi.org/10.1007/978-981-16-8150-9_14

b. Statistical Analysis of Stakeholders Perception on Adoption of AI/ML in Sustainable Agricultural Practices in Rural Development. In X. S. Yang, S. Sherratt, N. Dey, & A. (. Joshi (Ed.), Proceedings of Ninth International Congress on Information and Communication Technology. ICICT 2024 2024. Lecture Notes in Networks and Systems. 1003. Springer, Singapore. doi:https://doi.org/10.1007/978-981-97-3302-6_11

Response: Thank you for your valuable suggestion. The recommended references were included and cited in the resubmitted manuscript.

1.3 The problem that author is trying to address in this paper is not clear yet, (clearer motivation for study is required)

Response: We apologize for the lack of clarity in terms of the problem. To address this comment, we have further modified the motivation part:

 “In this work, we systematically address the challenge of recovering spatial temperature information from line-of-sight emission spectra measurements using data-driven modelling approaches. We first explore multiple feature engineering methodologies, and combine them with various machine learning (ML) algorithms. In parallel, we explore end-to-end deep learning (DL) algorithms for temperature distribution reconstruction purposes.” is revised to “The primary objective of this study is to develop and evaluate a comprehensive framework for spatially resolved temperature estimation from emission spectra, integrating feature engineering and ML approaches. By addressing the limitations of conventional Deep Learning (DL) and exploring the potential of hybrid methodologies, this research aims to enhance predictive accuracy and reliability, ultimately contributing to advancements in various applications ranging from industrial combustion monitoring to environmental assessments.”

1.4 Add the contribution of this study and study organization towards end of Introduction Section

Response: We would like to thank the reviewer for their comment. A dedicated paragraph which summarizes the key contributions of the study, and highlights the innovative aspects of the research has been included, i.e., 

“The contributions of the work are as follows:

1.To the best of the authors' knowledge, this is the first work which systematically explores and compares the use of feature engineering and machine learning vs deep learning algorithms for spatially resolved temperature measurement estimation.

2. The top performing method involves a combination of signal representation-based feature extraction, and blending-based ensemble machine learning, achieving cutting-edge performance, which outperforms state-of-the-art approaches and deep learning algorithms. 

3. It is demonstrated that the proposed approach is capable of accurately providing spatially resolved temperature estimates using lower resolution ES spectra, compared to published work, which offers an important cost advantage for real-world applications.”

Moreover, an outline of the manuscript structure was included at the end of the introduction: 

“The manuscript is organized as follows: Section 2 provides a detailed overview of the related works and the state-of-the-art methodologies in emission spectroscopy and temperature estimation. Section 3 outlines the methodology, including the data collection process, feature extraction techniques, and the machine learning models employed. In Section 4, the results of the experiments are presented, followed by a discussion of the findings in Section 5. Finally, Section 6 concludes the study and outlines future research directions.”

1.5 In 4-5- lines, authors should summarize the literature gaps identified before starting the methodology (Kindly relate your work in the context of existing work)

Response: We appreciate the reviewer’s comment. To address this, we have further modified the related work section and include a table to summarize and compare all these methods. Moreover, as suggested by the reviewer, we have included a paragraph summarizing the gaps in the existing literature on temperature estimation from emission spectroscopy: 

“Despite various advancements in both optimization-based and machine learning methods for temperature estimation from emission spectroscopy, significant gaps remain. Optimization-based approaches are often highly sensitive to initial conditions and hyperparameters, limiting their robustness in complex scenarios. Machine learning models, while promising, have primarily focused on simple temperature distributions and have struggled to generalize to more complex profiles due to insufficient feature engineering. Furthermore, few studies have explored the role of advanced feature extraction techniques to enhance model performance, especially in handling non-uniform temperature fields. This study addresses these gaps by incorporating comprehensive feature engineering with machine learning to improve accuracy in complex temperature estimations.”

This streamlined version clarifies the identified literature gaps and emphasizes the study's contribution.

1.6 Author should re-check the correctness of equations especially equations 8 and 14

Response: The equations have been re-checked, thank you.

1.7 Services of language expert are required

Response: The manuscript underwent review by a native English speaker and language issues were rectified.

1.8 Image quality needs to be improved

Response: All figures were reproduced at higher resolution (300 DPI) and adjusted labels and legends for better visibility.

1.9 Ensure that all Figures are properly labeled and referenced

Response: Verified and corrected figure labels and cross-referenced all figures in the text.

1.10 Authors should include the future work of this study

Response: Thank you for your valuable suggestions. In response, we have included a future work section at the end of the manuscript. The future directions for this study are as follows:

“As for future work, there are several aspects could be pursued.

 1. Data generation. More diverse temperature profile datasets could be generated, so as to further expand the application scenarios and assess the limitations of the proposed method. 

 2. Feature engineering and learning. More detailed hyper-parameter search, such as window size selection, and feature engineering methods, such as other dimensionality reduction methods, could be explored, for better performance. In addition to current feature engineering technologies, advanced feature learning technologies such as contrastive learning and self-supervised learning could be further explored, to extract more effective features for better performance.

 3. Incorporating hybrid machine learning approaches, where feature engineering is used to augment deep learning models, has the potential to significantly enhance prediction accuracy for temperature estimation from emission spectra. Hybrid methodologies capitalize on the strengths of both classical feature extraction techniques and modern deep learning architectures. By integrating domain knowledge through feature engineering with the learning capabilities of deep learning models, it is possible to create more robust and accurate predictive models.”

Reviewer #2: 

2.1 The title is clear and concise. However, consider specifying in the title that you compare different data-driven models, as this may better capture the comprehensive scope of your research.

Response: Thank you for your valuable comment, we have revised the title to: “Comparative Analysis of Data-Driven Models for Spatially Resolved Thermometry Using Emission Spectroscopy”.

2.2 Your systematic exploration of data-driven models is well-structured. Ensure that the methodology section provides detailed explanations of the implementation of feature engineering techniques, including physics-guided transformations, polynomial approximation-based features, and PCA.

Response: Thank you for your kind suggestion, we checked the details of the descriptions of the feature engineering techniques employed in the study, and ensured that each technique is explained in the context of its relevance to the study.

2.3 The methodology, results, and conclusions are clearly presented. Consider adding more discussion on practical implications and potential applications of your findings.

Response: To address this comment, we expanded the discussion as recommended by the reviewer, as follows:

“The findings of this study can contribute to the advancement of both measurement accuracy and the practical application of these techniques in various industries and research domains.

1. Advancements in Temperature Estimation Accuracy: The high performance of the feature-engineered models, demonstrated by the low RMSE and high correlation coefficients, suggests that data-driven approaches can enhance the accuracy of temperature measurements in scenarios where conventional methods may struggle. For instance, in spatially resolved thermometry, where detailed temperature profiles are required, the ability to accurately capture temperature variations across a region can lead to better control in industrial processes such as combustion monitoring, materials processing, and chemical reactors. Improved accuracy can lead to better operational efficiency, energy savings, and enhanced safety, particularly in environments where precise temperature control is critical.

2. Applications in Industrial and Environmental Monitoring: The findings from this study can be directly applied to improve real-time monitoring systems in various industrial applications. In industrial combustion, for example, accurate temperature monitoring enables operators to optimize fuel consumption, reduce emissions, and improve overall process efficiency. The high accuracy of the models in estimating temperatures from emission spectra can also facilitate the development of more advanced, automated monitoring systems for factories, power plants, and industrial furnaces, reducing the need for manual interventions.

3. Technological Improvements in Emission Spectroscopy Systems: The integration of the proposed models into existing emission spectroscopy systems could lead to technological improvements in terms of spatial resolution and measurement efficiency. Current systems may benefit from incorporating machine learning algorithms capable of analyzing spectral data more efficiently and with greater accuracy. This could pave the way for next-generation spectroscopic equipment that not only measures emission spectra with high spatial resolution but also provides real-time temperature data with minimal manual intervention. More importantly, it does not affect current hardware design, but empower the spatial measurement capability with intelligent algorithm embedding. These advancements could be particularly beneficial in scenarios requiring continuous monitoring and spatial resolution such as manufacturing processes, environmental assessments, and energy production.”

2.4 It is important that the description of the CNN models includes details on their architecture, training process, and evaluation metrics to provide a complete understanding of their performance relative to classical methods.

Thank you very much for your valuable comment, as for the CNN models, we use the default settings of these models. To address this, we included references for interested readers to obtain further details in regards to their architectures. In addition to this, we provided additional details about the training process and described the evaluation metrics for the readers’ convenience as follows:

“During training, the Adam optimizer [47] was used, and a weight decay of 1e−4 was selected to alleviate overfitting. The learning rate was preliminarily determined by the method suggested in [48], however, it was further decreased as the selected deep learning models were prone to overfitting the training set, particularly, after removing the dropout and normalization operations, required for the regression application. Thus, the learning rate was varied and determined for the specific CNN architecture. In order to better navigate the training direction, the warm-up operation [49] was used to train the networks in the first ten epochs, during which, the learning rate was gradually increased from a tiny value to the selected learning rate. Such a soft activation of training supports the appropriate choice of training direction. During the normal training process, the cosine annealing learning rate schedule [49] was used, which helps to avoid falling into local minima.” 

2.5 The presentation of performance metrics (RMSE: 64.3, RE: 0.017, RRMSE: 0.025, R: 0.994) is effective in highlighting the superiority of the light blender model. Ensure that the implications of these metrics are discussed in the context of practical applications

Response: Thank you for your insightful comment. In response, we have added a section in the conclusion to discuss the practical implications of the performance metrics and their relevance to real-world applications, as shown in the response for comment 2.3.

2.6 The acknowledgment that end-to-end CNNs are competitive but not superior is crucial. Consider providing more details on the specific limitations of CNNs in this context.

Response: We expanded the CNN section of the Discussion. Here, we focused on articulating the limitations of CNNs compared to other methods used in the study, emphasizing why they may not always be the best choice for temperature estimation from emission spectra. The following paragraphs have been included:

“Despite the well-documented success of Convolutional Neural Networks in various data-driven applications such as image classification, object detection, and time-series forecasting, their performance in the context of spatially resolved temperature measurement using emission spectroscopy was found to be inferior to classical machine learning methods in this study. Several key reasons contribute to this outcome:

• Loss of Spectral Characteristics During Convolution: Spectral data is characterized by its peaks and troughs, which directly correlate with the physical properties of the gas mixture, such as concentration and temperature distribution. Convolutional filters, designed for translation invariance in images, do not inherently capture these spectral characteristics effectively. As a result, important spectral features such as peak intensities and their precise locations might be averaged out during convolution, leading to a loss of critical information necessary for accurate temperature estimation.

• Challenges with High-Dimensional Spectral Inputs: The high-dimensional nature of the spectral data (e.g., 6799 emission

---

## [Decision Letter · Decision Letter 1]

4 Dec 2024

PONE-D-24-35923R1Comparative Analysis of Data-Driven Models for Spatially Resolved Thermometry Using Emission SpectroscopyPLOS ONE

Dear Dr. Liatsis,

Thank you for submitting your manuscript to PLOS ONE. After careful consideration, we feel that it has merit but does not fully meet PLOS ONE’s publication criteria as it currently stands. Therefore, we invite you to submit a revised version of the manuscript that addresses the points raised during the review process.

We look forward to receiving your revised manuscript.

Kind regards,

Anurag Sinha, Ph.D

Academic Editor

PLOS ONE

Journal Requirements:

Additional Editor Comments :

Comments from PLOS Editorial Office: We note that one or more reviewers has recommended that you cite specific previously published works in an earlier round of revision. As always, we recommend that you please review and evaluate the requested works to determine whether they are relevant and should be cited. It is not a requirement to cite these works and you may remove them before the manuscript proceeds to publication. We appreciate your attention to this request.

Reviewers' comments:

Reviewer's Responses to Questions

**Comments to the Author**

1. If the authors have adequately addressed your comments raised in a previous round of review and you feel that this manuscript is now acceptable for publication, you may indicate that here to bypass the “Comments to the Author” section, enter your conflict of interest statement in the “Confidential to Editor” section, and submit your "Accept" recommendation.

Reviewer #1: All comments have been addressed

Reviewer #3: (No Response)

Reviewer #4: (No Response)

2. Is the manuscript technically sound, and do the data support the conclusions?

Reviewer #1: Yes

Reviewer #3: Yes

Reviewer #4: Yes

3. Has the statistical analysis been performed appropriately and rigorously? 

Reviewer #1: Yes

Reviewer #3: Yes

Reviewer #4: Yes

4. Have the authors made all data underlying the findings in their manuscript fully available?

Reviewer #1: Yes

Reviewer #3: Yes

Reviewer #4: Yes

5. Is the manuscript presented in an intelligible fashion and written in standard English?

Reviewer #1: Yes

Reviewer #3: Yes

Reviewer #4: Yes

6. Review Comments to the Author

Reviewer #1: The manuscript have been revised and it is more suitable for acceptance to the best of my Knowledge

Reviewer #3: The authors present a very profound research that implements various data-driven models in measuring temperature distributions in a spatially resolved manner using emission spectroscopy data. However, here are few suggestions mentioned below:

1. The related works section has only 5 previous works that have been discussed in this paper. Perhaps the authors should try to add a few more previous related works.

2. Add a table that abbreviates all the variables and formulas used in the equations used in the paper so it can be easier for the readers to follow along.

3. What are the specific reasons to choose such machine learning models for comparison

4. Explain the main difference between Convolutional Neural Networks and the machine learning models, perhaps provide a comparison among the results

5. Make sure to proofread the entire manuscript for any grammatical errors.

Reviewer #4: The authors have conducted a greatly in-depth research related to addressing the caveat that line-of-sight emission spectroscopy and made changes accordingly to the first revision comments. Here are few comments I'd like to provide:

- I'd suggest the authors to provide a small overview for each section and subsection instead of directly jumping between one section to subsection without any overview or introduction. For example, Section 4 not only jumps one subsection but two, it goes from primary section 4 to 4.1, and then directly to 4.1.1.

- The conclusion portion of the paper should be presented in a paragraph format to match the appropriate format of a research article

- Are there any specific reasons only the following five CNN models were used for comparison: VGG, Resnet, Inception, Shuffle net, Squeeze net?

- Have the authors thought about adding another training set in the blending method, and if so, do you believe it can improve the prediction accuracy?

7. PLOS authors have the option to publish the peer review history of their article (what does this mean?). If published, this will include your full peer review and any attached files.

Reviewer #1: **Yes: **Sunday Adeola Ajagbe

Reviewer #3: No

Reviewer #4: No

---

## [Author Response · Author response to Decision Letter 1]

8 Dec 2024

Response to the Journal/Academic Editor:

We would like to sincerely thank the reviewers in this second round of review for their insightful comments and constructive suggestions. Their thorough evaluation and thoughtful feedback have greatly contributed to improving the clarity, depth, and overall quality of the submitted manuscript. In the following sections, we provide a point-by-point response to all comments made by the reviewers.

Reviewer #1: The manuscript have been revised and it is more suitable for acceptance to the best of my Knowledge

Thank you very much for your valuable feedback and review.

Reviewer #3: The authors present a very profound research that implements various data-driven models in measuring temperature distributions in a spatially resolved manner using emission spectroscopy data. However, here are few suggestions mentioned below:

1. The related works section has only 5 previous works that have been discussed in this paper. Perhaps the authors should try to add a few more previous related works.

Thank you for your valuable suggestion. We have added more references for emission-spectroscopy-based temperature measurements, and reorganized the section of related work. the new added references are as follows:

5. Parameswaran T, Hughes R, Gogolek P, Hughes P. Gasification temperature measurement with flame emission spectroscopy. Fuel. 2014;134:579–587. 

6. Yubero C, García MC, Varo M, Martínez P. Gas temperature determination in microwave discharges at atmospheric pressure by using different Optical Emission Spectroscopy techniques. Spectrochimica Acta Part B: Atomic Spectroscopy. 2013;90:61–67. 

17. Park JH, Cho JH, Yoon JS, Song JH. Machine learning prediction of electron density and temperature from optical emission spectroscopy in nitrogen plasma. Coatings. 2021;11(10):1221. 

18. Kim D, Bong C, Im Sk, Bak MS. Simultaneous measurement of carbon emission and gas temperature via laser-induced breakdown spectroscopy coupled with machine learning. Optics Express. 2023;31(4):7032–7046. 

19. Yi Y, Kun D, Li R, Ni K, Ren W. Accurate temperature prediction with small absorption spectral data enabled by transfer machine learning. Optics Express. 2021;29(25):40699–40709. 

2. Add a table that abbreviates all the variables and formulas used in the equations used in the paper so it can be easier for the readers to follow along.

We agree with your recommendation that providing an abbreviation table for all symbols used in the paper would be convenient for readers to follow, however, it is notable that we do not find the available format in the journal template for this function. With this in mind, we provided detailed definitions of the features extracted in the revised Tables 2 and 3.

3. What are the specific reasons to choose such machine learning models for comparison

The selection of these machine learning models was guided by several key considerations, as follows:

Model Diversity: The chosen models represent a diverse range of machine learning approaches, covering major learning paradigms: 

• Multi-Layer Perceptron (MLP) as a neural network-based approach

• Gaussian Process Regression (GPR) for probabilistic regression

• Support Vector Regression (SVR) for non-linear regression with kernel methods

• Radial Basis Function Networks (RBFN) for function approximation

• Ensemble methods (Random Forests and Boosted Trees) for providing high prediction robustness and the capability to combine already trained models.

Complementary Strengths: 

• MLP can capture complex non-linear relationships through its hidden layers and globally parameterized learning style.

• GPR provides uncertainty quantification and handles small datasets well with non-parameterized learning style

• SVR excels in high-dimensional spaces and manages non-linear relationships

• RBFN is particularly effective for interpolation and function approximation with regional learning style.

• Ensemble methods (Random Forests and Boosted Trees) offer robust predictions by combining multiple weak learners, reducing overfitting and improving generalization

By employing this comprehensive suite of models, we aimed at providing a robust and multi-perspective approach to estimating temperature profiles, ensuring a thorough and systematic investigation of predictive capabilities across different machine learning technologies.

Accordingly, we added the following paragraph to accommodate the reviewer’s comment in the manuscript.

“To ensure a robust and comprehensive investigation of temperature profile estimation, a diverse suite of machine learning models was employed, each representing distinct learning paradigms with complementary strengths. The chosen methods include Multi-Layer Perceptron (MLP) [35] for capturing complex non-linear relationships, Gaussian Process Regression (GPR) for its probabilistic predictions and suitability in small datasets [36], Support Vector Regression (SVR) [37] for its efficiency in high-dimensional spaces, and Radial Basis Function Networks (RBFN) [38] for effective interpolation and localized function approximation. Additionally, ensemble methods [39], i.e., Random [40] and Boosted Trees [41] were utilized for their ability to combine multiple weak trees, offering robustness and improved generalization. In addition to the above ML models, a blending method is also employed to provide superior performance based on the aforementioned models' performance through a two-stage training process, as shown in Figure 6. The first stage models are the weak learners, i.e., the ML models used to tackle the problem individually. The second stage of the process is the blending model or meta learner, which provides the final estimates, using the primary estimates from the weak learners. The meta learner can be a linear (e.g., weighted average) or a nonlinear model (e.g., MLP). This diverse model selection ensures that complementary perspectives and capabilities are leveraged, enabling a systematic evaluation of the predictive performance across different machine learning approaches.”

In addition to this, we also accommodated a justification for the selection of the specific Deep Learning methodologies in the manuscript, as follows:

“The selection of VGG, Resnet, Inception, Shuffle Net, and Squeeze Net for comparison in this study was guided by their prominence in the field and their architectural diversity. VGG represents one of the earliest and most well-known convolutional neural network architectures, emphasizing deep stacking of layers for feature extraction. Resnet introduces the concept of residual learning through skip connections, which addresses the vanishing gradient problem in deep networks. Inception networks leverage multi-scale convolutional kernels, allowing them to extract features across different receptive fields, making them effective for varied data patterns. Shuffle Net and Squeeze Net, on the other hand, are lightweight architectures designed for computational efficiency, making them practical for scenarios with limited resources. These models were chosen as they collectively cover a wide spectrum of design philosophies, from classical architectures to modern lightweight designs, ensuring a comprehensive evaluation.”

4. Explain the main difference between Convolutional Neural Networks and the machine learning models, perhaps provide a comparison among the results.

In our research, we contrasted the use of Convolutional Neural Networks (CNNs) and traditional machine learning (ML) models according to their difference in feature engineering: classical ML algorithms inherently struggle with high-dimensional data due to the well-documented dimensionality curse. Consequently, feature engineering becomes essential to compressing high-dimensional sparse data into low-dimensional dense features that ML regressors can effectively utilize. CNNs, in contrast, employ convolution filters and gradient backpropagation to learn feature representations automatically, potentially mitigating the dimensionality challenge and reducing manual feature engineering workload. However, it is crucial to note that the CNN's design and convolution filter capabilities may not always guarantee optimal feature representation. In contrast, classical ML regressors offer the flexibility to employ diverse feature extraction and selection methodologies, as highlighted in our response to comment 3, enabling a more nuanced and potentially more accurate feature engineering approach.

Their difference in feature engineering and the corresponding effect on their performance has been discussed in the following narrative in the manuscript:

“Despite the well-documented success of Convolutional Neural Networks in various data-driven applications such as image classification, object detection, and time-series forecasting, their performance in the context of spatially resolved temperature measurement using emission spectroscopy was found to be inferior to the combination of advanced feature engineering and classical machine learning methods in this study. Several key reasons contribute to this outcome:

• Loss of Spectral Characteristics During Convolution: Spectral data is characterized by its peaks and troughs, which directly correlate with the physical properties of the gas mixture, such as concentration and temperature distribution. Convolutional filters, designed for translation invariance in images, do not inherently capture these spectral characteristics effectively. As a result, important spectral features such as peak intensities and their precise locations might be averaged out during convolution, leading to a loss of critical information necessary for accurate temperature estimation.

• Challenges with High-Dimensional Spectral Inputs: The high-dimensional nature of the spectral data (e.g., 6799 emission lines) poses a challenge for CNNs, which rely on down-sampling through pooling layers to reduce computational complexity. However, this reduction can lead to information loss, especially when the input spectra contain fine-grained details that are crucial in distinguishing subtle variations in temperature. In contrast, feature engineering methods, e.g., polynomial approximations and PCA, preserve this high-dimensional information more effectively by focusing on extracting relevant features without losing the global context of the spectral data.

• Insufficient Feature Representation Learning: CNNs tend to learn hierarchical features through multiple layers of convolutional filters, which is highly beneficial for image data. However, in the case of emission spectra, where the relevant features are not hierarchical in nature, the learned representations may not align with the true underlying patterns. As demonstrated in this study, feature engineering techniques such as physics-guided transformations and signal representation features provide a more direct representation of the spectral data, allowing classical machine learning models to effectively leverage these features.

• Feature Engineering with Domain-Specific Knowledge: The feature engineering techniques employed in this study incorporate domain-specific knowledge through physics-based transformations and signal representation features. They are explicitly designed to capture the relationships between temperature, species concentration, and spectral intensities. This contrasts to CNNs, which lack such domain-specific priors and rely entirely on the data to learn these relationships. As a result, CNN methods often require larger datasets and extensive training to match the performance of models utilizing engineered features.”

5. Make sure to proofread the entire manuscript for any grammatical errors.

Thank you for your recommendation. We would like to confirm that the entire manuscript has been thoroughly proofread and any typographical/grammatical and syntactical errors have been addressed in the revised version.

Reviewer #4: The authors have conducted a greatly in-depth research related to addressing the caveat that line-of-sight emission spectroscopy and made changes accordingly to the first revision comments. Here are few comments I'd like to provide:

1. I'd suggest the authors to provide a small overview for each section and subsection instead of directly jumping between one section to subsection without any overview or introduction. For example, Section 4 not only jumps one subsection but two, it goes from primary section 4 to 4.1, and then directly to 4.1.1.

Thank you for your valuable comment. As recommended by the reviewer, we added the necessary overview to better organize the sections, and improve the flow of the manuscript. The following additions to the manuscript were made:

Section 3:

“…Sec. 3.1 will introduce the formulation of the forward modeling in emission spectroscopy, followed by spectral acquisition. In Sec.3.2, the data-driven methodologies explored in this study will be introduced.”

Section 3.1:

“This section primarily covers the data preparation process for data-driven modeling. Sec. 3.1.1 discusses the development of the physical forward model for spectra generation from temperature and mole fraction profiles. In Sec. 3.1.2, the physical model is applied in synthesizing the spectral data for use in the data-driven system development and testing.”

Section 4:

“The data-driven methodologies introduced in Sec.3.2 were used in the experimental studies, and the corresponding results are provided below. Sections 4.1 and 4.1 report the performance of the combination of feature engineering and machine learning, and Convolutional Neural Networks, respectively. Sec.4.3 provides a performance comparison between the top performing model in our experiments and the state-of-the-art approaches, demonstrating the effectiveness of the proposed approach. Sources of potential errors and a critical analysis of the limitations of the proposed method are discussed in Sec.4.4.”

Section 4.1:

“The high dimensionality and sparsity of the raw spectral data may lead to divergence of ML models, which necessitates the use of feature engineering to extract and select valuable features for information representation. Motivated by the above observation, Sec.4.1.1 demonstrates and discusses the performance of the proposed systematic feature engineering methodology, followed by the presentation and discussion of the optimal feature groups for machine learning modeling, and their corresponding performances in Sec.4.1.2.”

2. The conclusion portion of the paper should be presented in a paragraph format to match the appropriate format of a research article

Thank you for your valuable comment, we modified the conclusion section to accommodate the reviewer’s comment as follows:

“In this study, the feasibility of using data-driven models to recover spatially resolved temperature distributions from line-of-sight emission spectroscopy data was systematically explored, focusing on two primary approaches, i.e., feature engineering with classical machine learning methods, and end-to-end convolutional neural networks. The research yielded several significant findings. First and foremost, the combination of physics-guided transformation, polynomial approximation-based features, and Principal Component Analysis emerged as the most effective feature extraction methodology. Next, among fifteen traditional machine learning models, the light blender model demonstrated exceptional performance, achieving remarkable metrics including a root mean square error of 64.3, relative error of 0.017, relative root mean square error of 0.025, and an impressive R-value of 0.994. This result not only validates the feasibility of data-driven models in measuring nonuniform spatial temperature distributions but also highlights the substantial advantages of feature engineering.

While end-to-end Convolutional Neural Networks offered competitive performance compared to state-of-the-art techniques, they could not match the performance of feature engineering and machine learning models. This finding underscores the critical role of carefully engineered features in enhancing predictive accuracy for spectroscopic temperature estimation.

The implications of this research extend beyond methodological advancements. The high-performing models

---

## [Editor Report · Decision Letter 2]

3 Jan 2025

Comparative Analysis of Data-Driven Models for Spatially Resolved Thermometry Using Emission Spectroscopy

PONE-D-24-35923R2

Dear Dr. Liatsis,

We’re pleased to inform you that your manuscript has been judged scientifically suitable for publication and will be formally accepted for publication once it meets all outstanding technical requirements.

Kind regards,

Anurag Sinha, Ph.D

Academic Editor

PLOS ONE
---

## [Editor Report · Acceptance letter]

13 Jan 2025

PONE-D-24-35923R2 

PLOS ONE

Dear Dr. Liatsis, 

I'm pleased to inform you that your manuscript has been deemed suitable for publication in PLOS ONE. Congratulations! Your manuscript is now being handed over to our production team.

Kind regards, 

on behalf of

Mr. Anurag Sinha 

Academic Editor

PLOS ONE